# Confidence Calibration in Source-Free Domain Adaptation based on Pseudo-Labels

## Abstract

In this study, we explore the setting of source-free domain adaptation where access to labeled data from the source domain is restricted. We address the challenges associated with calibrating the prediction uncertainty of the adapted network solely based on unlabeled data from the target domain. To approximate the unknown true labels, we use pseudo-labels generated by the source model. Despite the high level of noise in pseudo-labels, our empirical analysis reveals that the network's accuracy computed using them closely matches the accuracy obtained with the true labels. Based on this observation, we propose a strategy for source-free confidence calibration. Our method is evaluated on standard domain adaptation benchmarks and achieves performance comparable to, or even better than, methods that require access to source data. Moreover, we significantly improve upon the state-of-the-art in source-free confidence calibration methods.

## 1 Introduction

Modern machine-learning systems are increasingly deployed in safety-critical applications such as medical diagnosis, autonomous driving, and financial risk assessment. In these domains, well-calibrated confidence estimates are as crucial as raw predictive accuracy: overconfident errors may lead to unsafe decisions, while underconfident predictions reduce trust in otherwise reliable models. Although post-hoc calibration techniques such as temperature scaling (Guo et al., 2017) have proven effective in supervised settings, their assumptions collapse under distribution shift, where the deployment domain differs from the training distribution (Ovadia et al., 2019).

A particularly challenging scenario arises in Source-Free Domain Adaptation (SFDA), where a pre-trained source model must be adapted to an unlabeled target domain without access to labeled or source data due to privacy, legal, or proprietary constraints. While SFDA methods have made significant progress using self-training and pseudo-labeling strategies to overcome the domain gap (see e.g. (Liang et al., 2020; Chen et al., 2022; Karim et al., 2023; Yi et al., 2023; Zhang et al., 2023)), the complementary problem of confidence calibration after adaptation remains largely unsolved. Existing calibration approaches typically assume either labeled target data or access to labeled source samples for importance weighting (Park et al., 2020; Wang et al., 2020) or distribution matching (Pampari & Ermon, 2020). These assumptions are incompatible with SFDA. Thus, despite the central role of pseudo-labels in adaptation, their role in supporting calibration has not yet been systematically explored.

In this work, we close this gap by showing that pseudo-labels, though noisy, can still serve as reliable surrogates for accuracy estimation in SFDA. Our key empirical observation is that the network accuracy computed using pseudo-labels closely matches the accuracy obtained with true labels, even under substantial noise. Building on this insight, we introduce Pseudo-Label Confidence Calibration (PLCC), a lightweight and principled algorithm that leverages pseudo-labels for bin-wise accuracy estimation and applies temperature scaling to calibrate the adapted model. Across four standard SFDA benchmarks, PLCC achieves state-of-the-art calibration performance, consistently outperforming both source-dependent and source-free baselines, while remaining simple and computationally efficient. Beyond advancing SFDA reliability, our results highlight a broader lesson: the same pseudo-labeling mechanisms that enable adaptation can also enable trustworthy post-adaptation calibration.

Our contributions are threefold:

- **Empirical insight.** We show that pseudo-labels, despite their inherent noise, yield accuracy estimates that closely match those obtained with true labels. This observation provides the foundation for calibration in the absence of ground-truth supervision.

- **A novel calibration method.** We introduce Pseudo-Label Confidence Calibration (PLCC), a simple yet effective algorithm that leverages enhanced pseudo-labels for bin-wise accuracy estimation and applies temperature scaling directly on the unlabeled target domain.

- **Extensive empirical validation.** We evaluate PLCC on four standard SFDA benchmarks (Office-Home, Office-31, VisDA, and DomainNet), demonstrating consistent state-of-the-art calibration performance, surpassing both source-free and source-dependent baselines while incurring negligible computational cost.

## 2 BACKGROUND AND RELATED WORK

**Confidence calibration.** Consider a network designed to classify an input $x$ into one of $k$ categories. Although the output of the softmax layer has the mathematical structure of a probability distribution, it does not always reflect the true posterior distribution of the classes. Networks often exhibit overconfidence in their predictions (Guo et al., 2017; Lakshminarayanan et al., 2017; Hein et al., 2019). The network prediction is defined as $\hat{y} = \arg\max_i p(y = i|x)$ and its confidence is $\hat{p} = \max_i p(y = i|x)$. A network is calibrated if $p(\hat{y} = y|\hat{p} = p) = p$, for all $p \in [0, 1]$. The Expected Calibration Error (ECE) Naeini et al. (2015) is a practical way to measure model calibration. It involves partitioning confidence values of a given set into $M$ equal-size bins, where $B_m$ is the index set of samples falling into the $m$-th bin. The ECE measure calculates the weighted average of the accuracy-confidence difference across all bins:

$$\text{ECE} = \sum_{m=1}^{M} \frac{|B_m|}{n} |A_m - C_m| \tag{1}$$

such that $A_m$ and $C_m$ are the average accuracy and confidence at the $m$-th bin and $n$ is the number of samples used to compute the ECE measure. Adaptive ECE (adaECE) Nguyen & O'Connor (2015) is a more robust variant of ECE where the bin sizes are calculated to ensure an even distribution of samples across the bins ensuring that each bin contains $1/M$ of the data points with similar confidence values.

Various calibration methods have been developed. Network calibration can be integrated with training (see e.g. (Mukhoti et al., 2020; Müller et al., 2019; Zhang et al., 2022)), or it can be performed post-hoc using scaling methods like Platt scaling (Platt et al., 1999), isotonic regression (Zadrozny & Elkan, 2002). Temperature Scaling (TS) is a widely used and highly effective method for calibrating the output distribution of a classification network (Guo et al., 2017). This technique involves using a single parameter, $T$, to rescale the logit scores before computing the class distribution.

$$p_T(y = i|x) = \frac{\exp(z_i/T)}{\sum_{j=1}^{k} \exp(z_j/T)}, \quad i = 1, \dots, k \tag{2}$$

s.t. $z_1, ..., z_k$ are the logit values obtained by applying the network to input vector $x$. The optimal $T$ can be found by minimizing either the ECE or the adaECE measures for the held-out validation dataset. because the sample is evenly split across the bins.

**Calibration under domain shift.** Direct calibration using data from the target domain is challenging in the absence of ground-truth labels. Almost all calibration methods heavily rely on the availability of labeled data from the source domain. Several studies (Salvador et al., 2021; Tomani et al., 2021) proposed modifying the calibration set to represent a generic distribution shift. Other studies (Park et al., 2020; Wang et al., 2020; Pampari & Ermon, 2020) applied Importance Weighting (IW) by assigning higher weights to source examples that resemble those in the target domain. These methods are not applicable when access to source domain data is restricted. The only SFDA calibration method we are aware of is PseudoCal (Hu et al., 2023) which is based on creating a mixture of images from the target domain.

**SFDA based on Pseudo Labels.** Most SFDA methods use pseudo-labels generated based on the source model predictions to adapt the source model to the target domain. To generate more accurate

pseudo-labels, we can use unsupervised techniques and self-training (Zhang et al., 2023; Liang et al., 2020). This involves utilizing the source model's predictions on target data along with a pre-trained strong feature extractor $f_p$ (Swin-B) (Liu et al., 2021), to create centroids for each class. Cosine distance is then used to assign each example to its nearest centroid. We denote the obtained labels Enhanced Pseudo Labels (EPL). Even the enhanced pseudo-labels tend to be very noisy for two reasons: (1) they are derived from predictions made by deep learning models and (2) the model generating these pseudo-labels is applied to a different domain from the one it was trained on. Many SFDA methods are based on explicitly handling the inaccuracy of the pseudo-labels by recasting the SFDA problem into the problem of learning with noisy labels (Chen et al., 2022; Karim et al., 2023; Kumar et al., 2023; Litrico et al., 2023; Yi et al., 2023; Diamant et al., 2024). These SFDA methods consider the pseudo-labels as noisy labels and apply standard methods for learning with noisy labels, e.g. Sukhbaatar et al. (2015); Xiao et al. (2015); Goldberger & Ben-Reuven (2017); Zhang et al. (2021); Li et al. (2021); Lin et al. (2023).

## 3 CONFIDENCE CALIBRATION BASED ON PSEUDO-LABELS

**Problem Statement.** Consider an SFDA scenario with a $k$-way classification task. Let $g_s$ be a model that was trained on labeled data from the source domain and $g_t$ be the model adapted to the target domain. We are given unlabeled target domain samples $\{x_i\}_{i=1}^n$ and our goal is to calibrate the network confidence by minimizing the adaECE score. Estimating adaECE and optimizing the temperature parameter $T$ requires calculating bin-wise accuracy and confidence values. In the SFDA setup, we can still compute the network confidence for each sample. However, since we do not have labeled data from the target domain, we cannot compute the bin-wise average accuracy by solely using the unlabeled target domain data. We thus aim to find a temperature $T$ on the target domain without true labels.

Inspired by the successful pseudo-label based line of research for SFDA, a natural strategy for source-free confidence calibration is to consider pseudo-labels as noisy labels and recasting the source-free calibration problem into the problem of confidence calibration with noisy labels (see e.g. Noisy Temperature Scaling (NTS) (Penso et al., 2024)). All the noisy-label algorithms — whether for learning or calibration — mentioned above, are based on the assumption that conditioned on the true label, the noisy label and the input image are independent, i.e. $p(\tilde{y}|y, x) = p(\tilde{y}|y)$ such that $x$ is the input sample and $y$ and $\tilde{y}$ are the correct label and its noisy version. In our case of viewing pseudo-labels as a noisy version of the true labels, the label noise is strongly correlated with the image content. Actually, the pseudo-labels satisfy $p(\tilde{y}|y, x) = p(\tilde{y}|x)$ since they are solely a function of the image. Specifically, we expect a correlation between the correctness of the pseudo-label assigned to an image and the confidence of the target model in this image. This dependence between image content and pseudo-labels renders standard noisy-label calibration methods, such as NTS, ineffective for our calibration task. It is worth noting that although the SFDA methods discussed above also assume conditional independence between the features and the noisy labels, training procedures are generally less sensitive to this label noise assumption than calibration. This is because the highly non-linear structure of neural networks makes them more robust, whereas the temperature scaling process used for calibration is linear and thus more vulnerable to modeling inaccuracies.

In SFDA network training, pseudo-labels are inherently noisy and cannot be used directly. Therefore, it is necessary to explicitly address the label noise problem on a per-sample basis. The crux of our approach is that achieving a good accuracy estimation by directly using pseudo-labels instead of the true labels, does not require them to be noise-free. Noisy pseudo-labels can still be directly used for accuracy estimation. We only need the network *average* accuracy evaluated with the pseudo-labels to be similar to the one evaluated with the true labels.

Intuitively, we expect to observe incorrect pseudo-labels in cases where the learned features do not represent the image class well in the target domain, which results in both the true label $y$ and pseudo-label $\tilde{y}$ seeming plausible. In such cases, it can be also difficult for the adapted network to decide between $y$ and $\tilde{y}$. This implies that both $y$ and $\tilde{y}$ are reasonable labels. As a result, the adapted network tends to classify these examples as either $y$ or $\tilde{y}$ in nearly equal proportions. We can formalize this intuition in the following way:

$$p(\hat{y} = \tilde{y}|\tilde{y} \neq y) \approx p(\hat{y} = y|\tilde{y} \neq y). \tag{3}$$

---

**Algorithm 1** Pseudo-Label based Confidence Calibration (PLCC)

---

**Input**: Source model $g_s$, target model $g_t$, target held out dataset $\{x_i\}_{i=1}^n$ and pre-trained feature extractor $f_p$.

1: Calculate class centroids as a weighted average of the features $f_p(x)$:

$$c_j = \frac{\sum_i p(y_i = j|x_i) f_p(x_i)}{\sum_i p(y_i = j|x_i)}, \qquad j = 1, ..., k$$

where $p(y_i = j|x_i)$ is the class probability based on the source model $g_s$.

2: For each target instance $x_i$, generate an enhanced pseudo-label (EPL) $\tilde{y}_i$ based on its nearest centroid using the cosine distance:

$$\tilde{y}_i = \arg\min_j \cos(c_j, f_p(x_i)).$$

3: Compute the adapted network predictions and their confidence values on the target samples, $(\hat{y}_i, \hat{p}_i), i = 1, .., n$.

4: Estimate the bin-wise accuracy $\tilde{A}_i = \frac{1}{|B_i|} \sum_{t \in B_i} 1_{\{\hat{y}_t = \tilde{y}_t\}}$ at each bin.

5: Determine a temperature $\hat{T}$ that minimizes the adaECE score: $\hat{T} = \arg\min_T \sum_{i=1}^M |\tilde{A}_i - C_i(T)|$.

---

Since it is always true that $p(\hat{y} = \tilde{y}|\tilde{y} = y) = p(\hat{y} = y|\tilde{y} = y)$, we can also write (3) as:

$$p(\hat{y} = \tilde{y}) \approx p(\hat{y} = y). \tag{4}$$

Note that the probabilities in (4) are not conditioned on either the image or the true class; instead, they are defined as marginal distributions. Hence, the probabilistic assumption is very mild. This mild probabilistic assumption is sufficient to justify using pseudo-labels for calibration. The main novel observation of this study is the empirical validity of (4) that pseudo-labels, despite their inherent noise, yield accuracy estimates that closely match those obtained with true labels. We empirically validate this claim in Section 5.

The fact that the average accuracy computed using pseudo-labels approximates the true accuracy enables performing confidence calibration without access to ground-truth labels. Assuming that the pseudo-labels satisfy (4), we can use $n$ unlabeled samples from the target domain to estimate the network's (bin-wise) accuracy even if the pseudo-labels are very noisy:

$$\tilde{A} = \frac{1}{n} \sum_{i=1}^n 1_{\{\hat{y}_i = \tilde{y}_i\}} \approx \frac{1}{n} \sum_{i=1}^n 1_{\{\hat{y}_i = y_i\}} = A. \tag{5}$$

**SFDA Confidence Calibration.** Applying the bin-wise accuracy estimation on each confidence bin separately, yields an estimation $\tilde{A}_i = \frac{1}{|B_i|} \sum_{t \in B_i} 1_{\{\hat{y}_t = \tilde{y}_t\}}$ of the average accuracy at the $i$-th confidence bin $A_i = \frac{1}{|B_i|} \sum_{t \in B_i} 1_{\{\hat{y}_t = y_t\}}$. The bin-wise average confidence can be computed on the unlabeled target data. Thus, we can apply the Temperature Scaling (TS) calibration of the target model directly to the target domain data. We dub this calibration algorithm Pseudo-Label Confidence Calibration (PLCC). The PLCC procedure is summarized in Algorithm 1 and Fig. 1 presents a block diagram of the PLCC algorithm.

## 4 EXPERIMENTS

In this section, we evaluate the capabilities of PLCC to calibrate a network on a target domain after applying an SFDA procedure and assess its accuracy.

**Datasets.** We report experiments on the following standard domain adaptation benchmarks: Office-Home (Venkateswara et al., 2017), Office-31 (Saenko et al., 2010), VisDA (Peng et al., 2017), and DomainNet (Peng et al., 2019). Office-home is a dataset that contains 4 domains where each domain consists of 65 categories. The four domains are: Art (A) – artistic images in the form of sketches, paintings, ornamentation, etc.; Clipart (C) – a collection of clipart images; Product (P) – images

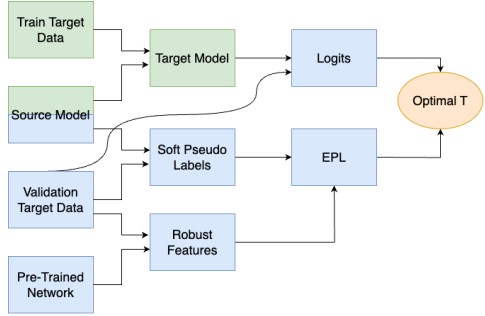

Figure 1: Diagram of the PLCC method: (green) Network adaptation process, (blue) the PLCC calibration of the adapted model, (orange) PLCC output.

of objects without a background and Real-World (R) – images of objects captured with a regular camera. Office-31 is a dataset that contains 31 object categories across three domains: Amazon (A), DSLR (D), and Webcam (W). These categories include common office items such as keyboards, file cabinets, and laptops. VisDA is a simulation-to-real dataset for domain adaptation with over 280,000 images across 12 categories. DomainNet is a large UDA dataset featuring common objects. The full dataset has 345 classes, but due to labeling noise in the complete version, we used two subsets: one with 126 classes (Zhang et al., 2023; Diamant et al., 2024) and the other with 40 classes (Tan et al., 2020; Diamant et al., 2024). We refer to these subsets as DomainNet126 and DomainNet40. Both subsets included four distinct domains: Clipart (C), Product (P), Real (R) and Sketch (S) images. Overall, we utilized 102 source-target pairs. There were 12 pairs from Office-Home, 1 from VisDA, 12 from DomainNet40, and 9 from DomainNet126. Since we applied three SFDA methods, the total number of source-target pairs was $(12+1+12+9) \times 3 = 102$.

**Compared calibration methods.** We compared our PLCC method against five baselines: (1) Uncalibrated: The adapted classifier used without any post-hoc calibration; (2) Source-TS: using the temperature learned on the source model with the source data to calibrate the target model, representing a scenario where this temperature was available in the adaptation process; (3) The NTS method (Penso et al., 2024) applied to the pseudo-labels we generated (only applicable for DCPL method, as it requires estimation of the noise transition matrix); (4) PseudoCal (Hu et al., 2023): an SFDA calibration method based on creating mixture of images from the target domain. (5) PLCC*: A variant of our PLCC method that uses less accurate pseudo-labels without applying EPL procedure (see Algorithm 1). It is presented as an ablation study to show the importance of applying the EPL procedure. Additionally, we implemented the following oracle results: (6) CPCS, (Park et al., 2020) and (7) TransCal, (Wang et al., 2020), both of which are importance-weighted UDA calibration methods. (8) UTDC (Penso & Goldberger, 2024) is a UDA calibration method that uses both the adapted model's source domain accuracy for each bin and an estimation of the target domain accuracy to calibrate the model. We used the UTDC* variant where the exact target domain accuracy of the adapted model was used instead of an estimation; (9) Target-TS, Temperature Scaling calibration (Guo et al., 2017) applied to the adapted network using the labeled validation set from the target domain.

**Implementation Details.** In our experiments, we employed three SFDA methods: DCPL (Diamant et al., 2024), SHOT (Jian Liang, 2020), and AaD (Yang et al., 2022) training all models to convergence using their official implementations. The three SFDA methods SHOT, AaD and DCPL represent three different approaches with respect to our generated pseudo-labels based calibration method. DCPL uses the same pseudo-labels without changing them during training, SHOT uses pseudo-labels that rely only on the source model and not on a strong pre-trained network (and also adapted during training) and AaD, does not use pseudo-labels during the target adaptation training. The CPCS and TransCal baselines were implemented with the code provided by the respective authors. To evaluate the UTDC*, NTS and PseudoCal methods, we also used the authors' provided code. For PseudoCal we set $\lambda = 0.65$ as recommended by the authors. Each dataset was tested using three different random seeds, and we report the average results. Due to the probabilistic nature of PseudoCal, TransCal and CPCS, we conducted 10 runs per seed and averaged the outcomes. For the calibration

Table 1: Adaptive ECE for top-1 predictions (in %) on **Office-Home**, using 15 bins (with the lowest in bold) across various SFDA classification tasks and methods with different calibration methods.

| SFDA | Type | Method | AC | AP | AR | CA | CP | CR | PA | PC | PR | RA | RC | RP | Avg |
|---|---|---|---|---|---|---|---|---|---|---|---|---|---|---|---|
| DCPL | Oracle | CPCS | 19.0 | 9.5 | 8.1 | 22.8 | 7.7 | 4.6 | 23.8 | 21.9 | 11.8 | 11.5 | 18.2 | 4.8 | 13.6 |
| | | TransCal | 17.1 | 7.4 | 5.6 | 13.2 | 6.9 | 4.5 | 10.8 | 17.4 | 7.3 | 9.8 | 16.7 | 3.8 | 10.0 |
| | | UTDC* | 9.1 | 6.4 | 3.7 | 6.6 | 6.8 | 4.4 | 10.3 | 8.9 | 6.8 | 11.1 | 8.8 | 3.6 | 7.2 |
| | | Target-TS | 7.4 | 6.0 | 3.4 | 6.1 | 5.3 | 3.5 | 7.5 | 8.6 | 4.7 | 6.0 | 7.9 | 3.2 | 5.8 |
| | | Uncalibrated | 23.5 | 10.3 | 6.1 | 10.6 | 8.8 | 7.5 | 10.8 | 26.7 | 7.3 | 9.8 | 23.0 | 6.4 | 12.6 |
| | | Source-TS | 28.6 | 13.1 | 10.0 | 16.6 | 11.4 | 11.6 | 16.6 | 32.0 | 11.3 | 15.6 | 27.1 | 8.9 | 16.9 |
| | | NTS | 21.5 | 12.6 | 10.1 | 17.3 | 12.0 | 9.8 | 19.7 | 23.7 | 10.3 | 17.9 | 19.8 | 8.1 | 15.2 |
| | | PseudoCal | 8.0 | 6.7 | 5.2 | 7.0 | 6.2 | 4.2 | 8.1 | 12.6 | 6.0 | 6.5 | 9.9 | 4.6 | 7.1 |
| | | PLCC* | 15.2 | 12.7 | 7.2 | 15.3 | 18.5 | 13.8 | 16.8 | 13.9 | 7.1 | 6.7 | 13.9 | 8.4 | 12.5 |
| | | PLCC | 8.4 | 7.1 | 3.6 | 6.4 | 6.1 | 4.4 | 7.7 | 11.5 | 5.2 | 7.0 | 10.7 | 3.8 | **6.8** |
| SHOT | Oracle | CPCS | 29.7 | 15.3 | 13.2 | 18.0 | 15.8 | 10.3 | 26.8 | 33.8 | 12.7 | 17.2 | 18.6 | 12.3 | 18.6 |
| | | TransCal | 28.4 | 11.3 | 8.1 | 18.2 | 9.8 | 8.5 | 11.6 | 29.8 | 6.8 | 9.2 | 18.9 | 8.4 | 14.1 |
| | | UTDC* | 6.5 | 7.3 | 4.5 | 7.8 | 6.7 | 5.0 | 8.6 | 8.6 | 7.1 | 11.9 | 7.4 | 5.1 | 7.2 |
| | | Target-TS | 5.8 | 6.9 | 3.9 | 6.6 | 5.9 | 4.3 | 7.7 | 7.4 | 4.2 | 6.7 | 6.9 | 3.9 | 5.9 |
| | | Uncalibrated | 28.6 | 13.6 | 8.1 | 12.4 | 13.3 | 10.9 | 11.6 | 30.6 | 6.8 | 9.2 | 28.1 | 8.4 | 15.1 |
| | | Source-TS | 36.2 | 17.9 | 13.5 | 19.7 | 17.1 | 16.0 | 18.6 | 38.7 | 12.6 | 16.4 | 33.6 | 12.0 | 21.0 |
| | | PseudoCal | 11.5 | 8.2 | 4.6 | 8.7 | 6.6 | 5.5 | 8.7 | 15.6 | 5.0 | 7.5 | 13.1 | 5.2 | 8.3 |
| | | PLCC* | 6.5 | 7.7 | 4.1 | 9.5 | 9.4 | 6.2 | 11.5 | 8.3 | 4.6 | 7.2 | 7.4 | 4.4 | 7.2 |
| | | PLCC | 7.4 | 8.4 | 5.3 | 7.9 | 6.7 | 4.9 | 8.8 | 11.3 | 4.8 | 9.1 | 10.3 | 4.5 | 7.4 |
| AaD | Oracle | CPCS | 29.2 | 11.3 | 11.5 | 25.9 | 14.4 | 6.5 | 25.1 | 24.7 | 12.0 | 15.4 | 22.0 | 7.1 | 17.1 |
| | | TransCal | 26.0 | 9.2 | 11.5 | 18.8 | 9.5 | 6.5 | 15.3 | 16.6 | 12.1 | 13.8 | 14.8 | 6.6 | 13.4 |
| | | UTDC* | 9.7 | 8.6 | 5.3 | 6.5 | 8.1 | 6.0 | 10.2 | 10.8 | 5.2 | 9.0 | 10.3 | 6.6 | 8.0 |
| | | Target-TS | 8.8 | 7.8 | 5.1 | 5.9 | 7.2 | 4.9 | 9.3 | 10.3 | 4.6 | 7.1 | 10.0 | 6.1 | 7.3 |
| | | Uncalibrated | 33.4 | 15.9 | 13.1 | 18.6 | 17.4 | 14.3 | 20.9 | 33.7 | 12.1 | 14.9 | 32.0 | 10.8 | 19.8 |
| | | Source-TS | 38.3 | 18.9 | 16.7 | 24.2 | 20.1 | 18.5 | 26.8 | 39.1 | 16.4 | 21.4 | 36.1 | 13.4 | 24.2 |
| | | PseudoCal | 14.5 | 10.1 | 5.7 | 8.4 | 8.5 | 6.0 | 12.0 | 17.3 | 4.8 | 8.0 | 13.9 | 7.0 | 9.7 |
| | | PLCC* | 10.7 | 9.7 | 5.6 | 9.9 | 11.7 | 8.7 | 12.2 | 14.8 | 5.1 | 8.3 | 11.7 | 7.6 | 9.7 |
| | | PLCC | 10.2 | 8.9 | 5.4 | 6.5 | 8.7 | 5.7 | 11.1 | 11.1 | 5.3 | 8.2 | 11.9 | 6.4 | **8.3** |

Table 2: Adaptive ECE for top-1 predictions (in %) on **VisDA**, using 15 bins (with the lowest in bold) across various SFDA methods with different calibration methods.

| Type | Method | DCPL | SHOT | AaD |
|---|---|---|---|---|
| Oracle | CPCS | 12.1 | 18.5 | 11.5 |
| | TransCal | 10.6 | 14.2 | 9.5 |
| | UTDC* | 5.0 | 3.4 | 3.1 |
| | Target-TS | 4.6 | 3.3 | 2.9 |
| | Uncalibrated | 13.7 | 15.5 | 12.2 |
| | Source-TS | 16.3 | 19.5 | 14.0 |
| | NTS | 9.8 | N/A | N/A |
| | PseudoCal | 9.8 | 5.7 | 10.1 |
| | PLCC* | 31.0 | 25.6 | 30.1 |
| | PLCC | **5.7** | **4.2** | **4.5** |

assessment, each target domain was split into 80% for training and 20% for validation. Adaptation was performed on the training set, and calibration was conducted on the validation set using adaptive ECE as the loss function. We report adaptive ECE results for the validation set. Additional ECE, Negative Log-Likelihood (NLL) (Hastie et al., 2009), Brier Score (BS) (Brier, 1950), and Static Calibration Error (SCE) (Nixon et al., 2019) results, along with adaptive ECE results for the Office-31 dataset (Saenko et al., 2010), are included in the Appendix. For reproducibility, we have made our code available [1].

**computational complexity.** Our method is extremely efficient. It does not involve any network training or parameter tuning. Note that pseudo-labeling does not introduce any computational overhead. It is performed by forwarding the target data through the source network under evaluation and considering the most probable prediction of that network as the label.

**Confidence Calibration Results.** Tables 1, 2, 4, and 3 present the calibration results for Office-Home, VisDA, DomainNet40, and DomainNet126, respectively. The findings show that PLCC outperformed the baseline methods in nearly all tasks. Additionally, compared to Oracle methods, PLCC consistently aligned with UTDC* and Target-TS. PLCC achieved good results for both SFDA methods that are based on pseudo-labels (DCPL and SHOT) and those which treat the SFDA problem as an unsupervised clustering problem (AaD). Furthermore, PLCC surpassed CPCS and TransCal in nearly all tasks, even though both methods have access to source domain data. Target domain calibration methods using labeled source data can generally be divided into two main approaches: (1) importance-weighting methods and (2) binwise average accuracy estimation methods. CPCS

---

[1] https:///anonymous.4open.science/r/SFCC-40E1

Table 3: Adaptive ECE for top-1 predictions (in %) on **DomainNet126**, using 15 bins (with the lowest in bold) across various SFDA classification tasks and methods with different calibration methods.

| SFDA | Type | Method | CR | CS | PC | PR | PS | RC | RS | SC | SR | Avg |
|---|---|---|---|---|---|---|---|---|---|---|---|---|---|
| DCPL | Oracle | CPCS | 11.5 | 20.6 | 14.3 | 8.6 | 20.8 | 14.2 | 16.1 | 9.8 | 8.0 | 13.8 |
| | | TransCal | 6.4 | 20.4 | 12.3 | 5.6 | 9.8 | 13.4 | 18.9 | 10.1 | 7.2 | 11.6 |
| | | UTDC* | 7.0 | 5.8 | 6.4 | 4.4 | 5.8 | 5.4 | 6.6 | 4.7 | 7.1 | 5.9 |
| | | Target-TS | 4.3 | 5.3 | 6.0 | 3.5 | 5.5 | 5.0 | 6.3 | 4.6 | 4.0 | 4.9 |
| | | Uncalibrated | 13.1 | 24.5 | 19.1 | 11.7 | 24.0 | 16.4 | 26.5 | 13.6 | 13.8 | 18.1 |
| | | Source-TS | 15.5 | 30.0 | 25.5 | 14.2 | 29.6 | 21.5 | 31.3 | 19.7 | 16.2 | 22.6 |
| | | NTS | 14.9 | 19.4 | 18.7 | 13.7 | 18.6 | 15.8 | 22.4 | 14.3 | 15.0 | 17.0 |
| | | PseudoCal | 4.7 | 6.1 | 7.4 | 4.1 | 6.0 | 6.8 | 7.1 | 5.2 | 4.6 | 5.8 |
| | | PLCC* | 11.2 | 7.2 | 11.5 | 6.1 | 13.6 | 7.5 | 8.1 | 7.4 | 11.7 | 9.4 |
| | | PLCC | 5.7 | 5.5 | 6.1 | 5.0 | 5.8 | 5.6 | 6.6 | 5.2 | 5.9 | **5.7** |
| SHOT | Oracle | CPCS | 12.7 | 19.3 | 21.8 | 11.7 | 21.8 | 17.5 | 16.1 | 11.9 | 6.7 | 15.5 |
| | | TransCal | 11.6 | 16.7 | 15.4 | 10.2 | 12.3 | 13.3 | 21.2 | 8.1 | 4.5 | 12.6 |
| | | UTDC* | 4.4 | 5.3 | 7.5 | 4.0 | 5.2 | 6.7 | 5.3 | 7.9 | 3.6 | 5.5 |
| | | Target-TS | 4.3 | 4.9 | 6.3 | 3.4 | 4.2 | 5.7 | 5.1 | 4.7 | 3.3 | 4.7 |
| | | Uncalibrated | 12.6 | 16.7 | 15.4 | 10.8 | 17.9 | 13.3 | 22.2 | 8.1 | 13.4 | 14.5 |
| | | Source-TS | 17.0 | 25.7 | 25.3 | 15.2 | 27.3 | 21.5 | 30.0 | 17.4 | 17.9 | 21.9 |
| | | PseudoCal | 5.3 | 6.6 | 10.0 | 4.2 | 5.7 | 6.3 | 5.5 | 5.1 | 4.4 | 5.9 |
| | | PLCC* | 7.8 | 4.9 | 8.3 | 3.7 | 9.6 | 6.1 | 5.8 | 5.8 | 7.5 | 6.6 |
| | | PLCC | 5.3 | 4.9 | 6.4 | 4.9 | 4.5 | 6.3 | 5.5 | 5.0 | 4.9 | **5.3** |
| AaD | Oracle | CPCS | 15.3 | 16.9 | 13.7 | 10.1 | 16.0 | 12.3 | 12.8 | 8.6 | 7.6 | 12.6 |
| | | TransCal | 10.0 | 20.7 | 13.8 | 7.5 | 6.5 | 11.1 | 14.2 | 9.2 | 8.0 | 11.2 |
| | | UTDC* | 7.7 | 5.4 | 7.6 | 5.3 | 5.1 | 6.7 | 6.6 | 4.1 | 7.0 | 6.2 |
| | | Target-TS | 6.7 | 4.9 | 7.1 | 4.7 | 4.9 | 5.4 | 5.5 | 3.6 | 5.7 | 5.4 |
| | | Uncalibrated | 17.0 | 22.9 | 21.4 | 12.3 | 22.4 | 15.8 | 26.5 | 9.7 | 15.8 | 18.2 |
| | | Source-TS | 19.8 | 29.7 | 29.1 | 15.2 | 29.6 | 22.3 | 32.6 | 17.3 | 18.8 | 23.8 |
| | | PseudoCal | 8.1 | 8.3 | 11.5 | 4.9 | 6.9 | 5.9 | 5.8 | 4.1 | 6.0 | **6.8** |
| | | PLCC* | 9.2 | 6.3 | 10.7 | 6.2 | 11.4 | 8.0 | 7.8 | 7.9 | 9.7 | 8.6 |
| | | PLCC | 7.0 | 6.4 | 7.6 | 5.2 | 7.3 | 8.1 | 7.9 | 7.3 | 6.0 | 7.0 |

Table 4: Adaptive ECE for top-1 predictions (in %) on **DomainNet40**, using 15 bins (with the lowest in bold) across various SFDA classification tasks and methods with different calibration methods.

| SFDA | Type | Method | CP | CR | CS | PC | PR | PS | RC | RP | RS | SC | SP | SR | Avg |
|---|---|---|---|---|---|---|---|---|---|---|---|---|---|---|---|
| DCPL | Oracle | CPCS | 13.7 | 2.0 | 14.0 | 15.2 | 5.8 | 9.0 | 10.3 | 4.9 | 15.1 | 8.7 | 10.1 | 14.3 | 10.3 |
| | | TransCal | 5.4 | 9.6 | 7.4 | 5.5 | 11.7 | 7.4 | 4.3 | 4.8 | 10.4 | 7.0 | 8.0 | 14.2 | 8.0 |
| | | UTDC* | 2.7 | 12.5 | 4.3 | 4.5 | 4.2 | 3.6 | 3.6 | 3.9 | 4.7 | 5.1 | 2.9 | 5.3 | 4.8 |
| | | Target-TS | 2.2 | 0.9 | 3.8 | 3.7 | 1.5 | 3.2 | 3.3 | 2.2 | 3.6 | 4.8 | 2.7 | 1.8 | 2.8 |
| | | Uncalibrated | 5.5 | 2.4 | 8.5 | 5.3 | 3.4 | 9.7 | 3.9 | 6.1 | 12.4 | 5.7 | 7.0 | 4.4 | 6.2 |
| | | Source-TS | 13.9 | 5.6 | 16.2 | 12.1 | 6.0 | 16.9 | 10.9 | 11.7 | 19.1 | 11.7 | 13.2 | 7.3 | 12.1 |
| | | NTS | 12.4 | 5.6 | 14.8 | 16.9 | 6.0 | 15.6 | 13.2 | 12.1 | 16.5 | 16.9 | 13.9 | 7.0 | 12.6 |
| | | PseudoCal | 3.0 | 6.5 | 4.8 | 5.3 | 6.1 | 3.6 | 7.1 | 4.8 | 5.1 | 6.3 | 4.3 | 5.9 | 5.2 |
| | | PLCC* | 18.9 | 13.1 | 12.8 | 11.1 | 5.6 | 6.4 | 7.0 | 2.3 | 8.3 | 11.5 | 10.1 | 10.5 | 9.8 |
| | | PLCC | 2.6 | 1.0 | 4.8 | 4.6 | 2.0 | 3.5 | 3.7 | 2.3 | 3.8 | 5.3 | 3.2 | 2.6 | **3.3** |
| SHOT | Oracle | CPCS | 13.0 | 6.8 | 13.5 | 20.4 | 3.7 | 5.6 | 6.3 | 4.0 | 9.0 | 13.0 | 15.9 | 14.9 | 10.5 |
| | | TransCal | 6.4 | 12.4 | 3.9 | 7.6 | 5.4 | 7.0 | 5.4 | 2.1 | 4.3 | 4.3 | 3.7 | 17.2 | 6.6 |
| | | UTDC* | 4.2 | 7.3 | 6.0 | 4.6 | 3.6 | 3.1 | 8.8 | 7.7 | 9.0 | 4.6 | 2.6 | 4.2 | 5.5 |
| | | Target-TS | 2.9 | 1.5 | 3.0 | 4.2 | 2.3 | 2.6 | 3.8 | 1.9 | 2.9 | 4.1 | 2.6 | 1.9 | 2.8 |
| | | Uncalibrated | 3.1 | 1.7 | 3.9 | 4.5 | 3.2 | 3.7 | 5.3 | 2.1 | 4.3 | 4.2 | 3.6 | 2.7 | 3.5 |
| | | Source-TS | 13.0 | 6.2 | 14.0 | 11.1 | 7.2 | 13.2 | 7.4 | 8.9 | 14.2 | 10.0 | 11.6 | 7.4 | 10.4 |
| | | PseudoCal | 3.3 | 3.7 | 3.8 | 4.8 | 3.7 | 3.5 | 7.2 | 3.5 | 4.9 | 5.3 | 3.0 | 3.9 | 4.2 |
| | | PLCC* | 14.8 | 11.5 | 9.3 | 6.3 | 3.0 | 4.3 | 4.7 | 3.8 | 6.2 | 8.4 | 8.3 | 8.7 | 7.4 |
| | | PLCC | 2.9 | 1.6 | 4.0 | 4.3 | 2.7 | 4.3 | 4.3 | 2.0 | 4.7 | 4.3 | 3.2 | 2.6 | **3.4** |
| AaD | Oracle | CPCS | 13.7 | 1.8 | 15.6 | 13.1 | 4.1 | 3.4 | 7.1 | 7.4 | 9.2 | 14.3 | 13.5 | 22.9 | 10.5 |
| | | TransCal | 10.5 | 10.7 | 4.5 | 14.4 | 4.2 | 6.3 | 4.3 | 3.8 | 6.8 | 6.2 | 10.4 | 19.4 | 8.5 |
| | | UTDC* | 2.7 | 8.4 | 3.2 | 4.1 | 3.2 | 2.4 | 9.2 | 6.7 | 6.6 | 3.6 | 2.9 | 6.3 | 4.9 |
| | | Target-TS | 2.1 | 1.3 | 2.6 | 3.6 | 1.1 | 1.7 | 3.0 | 2.3 | 2.2 | 3.5 | 2.8 | 1.1 | 2.3 |
| | | Uncalibrated | 3.3 | 2.9 | 4.5 | 3.9 | 3.7 | 5.5 | 3.4 | 3.8 | 6.6 | 3.7 | 4.2 | 2.4 | **4.0** |
| | | Source-TS | 13.0 | 6.3 | 13.5 | 10.2 | 6.1 | 13.7 | 8.6 | 10.4 | 14.9 | 8.8 | 10.4 | 5.5 | 10.1 |
| | | PseudoCal | 2.9 | 3.3 | 3.8 | 4.6 | 4.0 | 3.0 | 6.8 | 3.8 | 3.8 | 6.2 | 3.2 | 5.9 | 4.3 |
| | | PLCC* | 17.4 | 12.1 | 11.8 | 8.5 | 3.8 | 5.6 | 4.9 | 2.7 | 7.2 | 13.5 | 12.7 | 12.9 | 9.4 |
| | | PLCC | 3.8 | 1.3 | 6.8 | 5.6 | 1.2 | 6.6 | 5.8 | 3.5 | 5.6 | 6.6 | 4.8 | 1.5 | 4.4 |

and TransCal follow the first approach, while UTDC follows the second. Importance-weighting methods assume that source domain examples that are similar to target samples are more effective for calibrating target predictions, but this assumption often fails in practice (see (Penso & Goldberger, 2024)). On the other hand, methods that focus on estimating accuracy directly in the target domain tend to be more effective for calibration. PLCC is more closely aligned with this second category, as it estimates bin accuracy without relying on source domain data. As a result, our calibration outcomes were more consistent with the second approach and outperformed those from the first category.

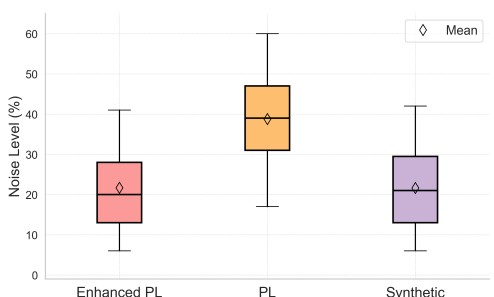 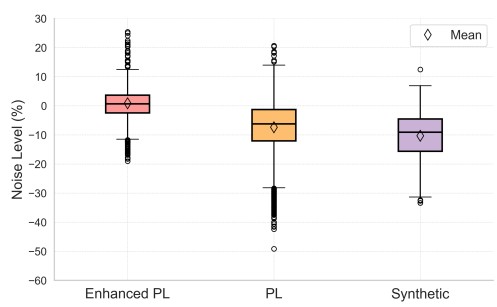

Figure 2: (left) Box plots showing the noise-level $p(y \neq \tilde{y})$ of different pseudo-labeling methods. (right) Box plots of the difference of estimating model accuracy using pseudo-labels instead of true labels.

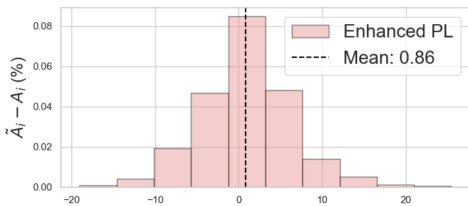

Figure 3: Histogram of $\tilde{A}_i - A_i(\%)$, the difference of estimating bin-wise model accuracy using pseudo-labels instead of true labels.

## 5 ANALYSIS

We next empirically validate that indeed pseudo-labels satisfies the noise behavior described in (4). Fig. 2a shows the noise level of the pseudo-labels across all 102 source-target pairs used in our experiments. We implemented three pseudo-label variants: (1) PL: pseudo-labels based solely on the source model's predictions. (2) Enhanced PL (see Algorithm 1). (3) Synthetic: synthetic noisy labels that were generated by a label-noise transition matrix computed from the conditional statistics $p(\tilde{y}|y)$ of the enhanced pseudo-labels given the true labels. We can see that enhancing the pseudo-labels reduces the average label noise level from 40% to 20% which is still high. The way the synthetic noise was created implies that it has the same noise level as the Enhanced PL. For each source-target domain pair and each confidence bin, we calculated the difference between estimated and true bin-wise accuracy: $\tilde{A}_i - A_i$. Fig. 2b presents box plots of this value across all the bins of the source-target pairs in our experiments. We can see that even though the noise level of the enhanced pseudo-labels is 20%, when using them to estimate the network accuracy, the error between the true accuracy and EPL-based estimated accuracy is close to zero. In contrast, in the case of synthetic label-noise, as expected, the estimated accuracy is consistently and significantly worse than the true accuracy. Fig. 3 shows the same information as Fig. 2b for the case of enhanced PL in a histogram

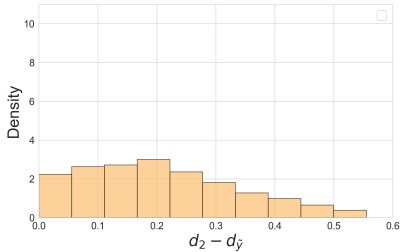 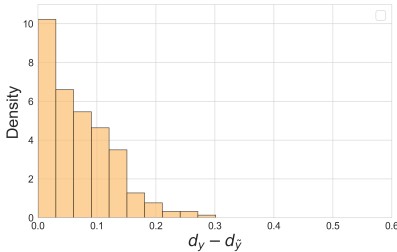

Figure 4: (a) Histogram of $d_2 - d_{\tilde{y}}$ in case of correct EPLs. (b) Histogram of $d_y - d_{\tilde{y}}$ in case of incorrect EPLs. Results are demonstrated on DomainNet40, (Sketch-Product).

format. Figs. 2 and 3 provide empirical evidence supporting the claim that model accuracy can be estimated directly from pseudo-labels. This justifies estimating the bin-wise model accuracy based on the pseudo-labels.

We next analyze the process of computing the enhanced pseudo-labels. The EPL $\tilde{y}$ is selected as the class of the nearest centroid to the embedding $f_p(x)$, i.e., $\tilde{y} = \arg\min_j \cos(c_j, f_p(x))$. The construction of the class-based centroids is described in Algorithm 1. Let $d_{\tilde{y}}(x)$, $d_2(x)$, and $d_y(x)$ be the cosine distances of $f_p(x)$ to the nearest class centroid, the second closest centroid, and the centroid of the true label, respectively. Fig. 4 illustrates the histogram of $d_2(x) - d_{\tilde{y}}(x)$ for correct EPLs, and the histogram of $d_y(x) - d_{\tilde{y}}(x)$ for incorrect EPLs. We can see in Fig. 4a that when the EPL is correct, it is usually a simple case and the correct label is evident from the image. In contrast, Fig. 4b shows that for incorrect EPLs, the average distance difference $d_y(x) - d_{\tilde{y}}(x)$ is small, suggesting that in this case the true and enhanced pseudo labels are similarly plausible. This is exactly the statement of (4).

# 6 CONCLUSIONS

We introduced a unified strategy for confidence calibration in source-free domain adaptation (SFDA), relying solely on unlabeled target domain data. Central to our method is a probabilistic noise model for pseudo-labels, which enables reliable accuracy estimation and calibration despite significant label noise. Our PLCC approach for confidence calibration—achieves new state-of-the-art performance on multiple domain adaptation benchmarks, outperforming existing source-free and source-dependent baselines. This method is not only effective but also simple and efficient, making them practical for real-world deployment where source data is unavailable. A notable insight from our work is that the same pseudo-labeling mechanisms used for model adaptation can also support downstream estimation and calibration tasks. This suggests a broader applicability of self-supervised techniques in evaluating and calibrating models post-adaptation. While our method performs well under moderate domain shifts, we observe limitations when the domain gap is too large, as pseudo-labels become unreliable. Addressing this limitation is a promising direction for future research.

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

# A APPENDIX

This appendix offers additional results not included in the main paper, accuracy levels of the SFDA methods, empirical evidence demonstrating that varying bin numbers do not impact the superiority of our method, and additional analysis that was not included in the main paper.

## A.1 ADDITIONAL EXPERIMENTAL RESULTS

In this section we provided additional adaECE results on Office-31 dataset (Saenko et al., 2010), which contains 31 object categories across three domains: Amazon (A), DSLR (D), and Webcam (W). These categories include common office items such as keyboards, file cabinets, and laptops. In this case we use adaECE for optimization and evaluating the results.

Additionally, we provide results for various calibration losses, including Expected Calibration Error (ECE), Negative Log-Likelihood (NLL), Brier Score (BS), and Static Calibration Error (SCE). In this context, all calibration methods used ECE optimization to find the optimal temperature, except for the UTDC* method. UTDC* optimized the adaECE metric but was still evaluated using ECE, NLL, BS, and SCE scores. This approach was chosen because UTDC* can experience a significant performance drop if the bin sizes are not roughly equal.

### A.1.1 ADAECE RESULTS

Table 5 presents adaECE score on Office-31 (Saenko et al., 2010) dataset, in this case PLCC algorithm outperforms all other calibration methods.

Table 5: Adaptive ECE for top-1 predictions (in %) on **Office-31**, using 15 bins (with the lowest in bold) across various SFDA classification tasks and methods with different calibration methods.

| SFDA | Type | Method | AD | AW | DA | DW | WA | WD | Avg |
|------|------|--------|-----|------|------|-----|------|-----|-----|
| DCPL | Oracle | CPCS | 3.0 | 5.9 | 16.5 | 1.1 | 17.7 | 0.7 | 7.5 |
| | | TransCal | 2.9 | 26.1 | 13.1 | 2.2 | 12.6 | 2.4 | 9.9 |
| | | UTDC* | 3.7 | 3.4 | 6.2 | 1.7 | 5.6 | 0.9 | 3.6 |
| | | Target-TS | 1.7 | 2.1 | 4.2 | 1.3 | 5.0 | 0.9 | 2.5 |
| | | Uncalibrated | 2.8 | 2.5 | 13.1 | 2.2 | 14.3 | 2.4 | 6.2 |
| | | Source-TS | 2.1 | 3.8 | 16.3 | 1.7 | 17.2 | 1.0 | 7.0 |
| | | NTS | 2.5 | 3.6 | 13.8 | 1.7 | 14.3 | 0.9 | 6.1 |
| | | PLCC* | 11.6 | 16.8 | 12.5 | 2.0 | 8.8 | 0.9 | 8.8 |
| | | PLCC | 2.0 | 4.3 | 6.2 | 1.8 | 8.0 | 0.9 | **3.9** |
| SHOT | Oracle | CPCS | 5.0 | 11.7 | 17.4 | 1.4 | 21.1 | 0.6 | 9.5 |
| | | TransCal | 4.9 | 17.5 | 11.5 | 4.8 | 14.7 | 5.2 | 9.8 |
| | | UTDC* | 4.3 | 5.2 | 6.5 | 2.0 | 5.6 | 0.9 | 4.1 |
| | | Target-TS | 3.2 | 4.5 | 4.9 | 1.7 | 4.5 | 0.9 | 3.3 |
| | | Uncalibrated | 4.9 | 5.2 | 11.5 | 4.8 | 14.7 | 5.2 | 7.7 |
| | | Source-TS | 4.1 | 7.6 | 17.1 | 1.8 | 19.9 | 1.1 | 8.6 |
| | | PLCC* | 8.6 | 11.7 | 11.3 | 2.8 | 5.0 | 0.9 | 6.7 |
| | | PLCC | 3.2 | 5.2 | 7.0 | 1.5 | 8.5 | 0.9 | **4.4** |
| AaD | Oracle | CPCS | 6.7 | 4.5 | 20.4 | 1.3 | 17.2 | 0.6 | 8.4 |
| | | TransCal | 3.3 | 17.9 | 15.7 | 2.6 | 15.4 | 2.4 | 9.5 |
| | | UTDC* | 3.4 | 3.8 | 6.8 | 1.5 | 5.3 | 0.9 | 3.6 |
| | | Target-TS | 3.3 | 3.1 | 5.9 | 1.2 | 4.4 | 0.9 | 3.1 |
| | | Uncalibrated | 3.3 | 3.1 | 15.7 | 2.6 | 15.4 | 2.4 | 7.1 |
| | | Source-TS | 5.1 | 4.7 | 19.0 | 1.6 | 18.7 | 0.9 | 8.3 |
| | | PLCC* | 9.4 | 14.7 | 10.4 | 2.9 | 6.7 | 0.9 | 7.5 |
| | | PLCC | 4.2 | 5.5 | 7.3 | 1.9 | 7.3 | 0.9 | **4.5** |

### A.1.2 ECE Results

Tables 6, 7, 8, and 9 provide the ECE calibration results for Office-Home, VisDA, DomainNet40, and DomainNet126, respectively.

Table 6: ECE for top-1 predictions (in %) on **Office-Home**, using 15 bins (with the lowest in bold) across various SFDA classification tasks and methods with different calibration methods.

| SFDA | Type | Method | AC | AP | AR | CA | CP | CR | PA | PC | PR | RA | RC | RP | Avg |
|------|------|--------|----|----|----|----|----|----|----|----|----|----|----|----|-----|
| DCPL | Oracle | CPCS | 18.5 | 8.7 | 7.0 | 18.9 | 6.6 | 4.5 | 20.8 | 20.6 | 11.0 | 8.1 | 18.8 | 3.6 | 12.3 |
| | | TransCal | 15.1 | 5.8 | 4.3 | 12.7 | 5.0 | 4.8 | 8.7 | 16.9 | 6.8 | 6.8 | 15.5 | 2.9 | 8.8 |
| | | UTDC* | 8.5 | 4.9 | 3.4 | 5.2 | 4.2 | 3.8 | 7.9 | 8.6 | 5.8 | 7.3 | 8.7 | 2.8 | 5.9 |
| | | Target-TS | 7.6 | 3.9 | 2.4 | 3.9 | 3.5 | 3.0 | 5.5 | 8.1 | 3.3 | 2.9 | 7.5 | 2.1 | 4.5 |
| | | Uncalibrated | 22.2 | 8.9 | 5.4 | 7.5 | 7.7 | 6.5 | 8.7 | 25.0 | 6.8 | 6.8 | 22.1 | 4.6 | 11.0 |
| | | Source-TS | 24.2 | 10.4 | 8.3 | 11.7 | 9.2 | 8.3 | 10.9 | 27.8 | 9.1 | 10.3 | 24.0 | 6.2 | 13.4 |
| | | NTS | 21.2 | 8.4 | 5.3 | 9.8 | 7.6 | 6.0 | 7.8 | 23.2 | 7.9 | 8.3 | 20.4 | 4.7 | 10.9 |
| | | PseudoCal | 8.7 | 5.1 | 4.1 | 5.0 | 4.4 | 4.0 | 6.6 | 13.0 | 4.5 | 4.2 | 9.6 | 3.2 | 6.0 |
| | | PLCC* | 14.5 | 11.7 | 6.6 | 16.7 | 19.0 | 13.8 | 18.0 | 12.7 | 4.3 | 3.4 | 14.2 | 6.4 | 11.8 |
| | | PLCC | 8.1 | 5.2 | 3.1 | 4.6 | 4.6 | 3.5 | 6.0 | 10.9 | 4.1 | 5.1 | 11.0 | 2.5 | **5.7** |
| SHOT | Oracle | CPCS | 29.1 | 13.7 | 11.6 | 14.7 | 13.8 | 9.3 | 21.5 | 32.7 | 11.8 | 12.8 | 18.9 | 9.9 | 16.7 |
| | | TransCal | 26.8 | 9.4 | 7.5 | 16.9 | 9.2 | 8.4 | 9.3 | 28.6 | 6.5 | 7.2 | 18.8 | 7.0 | 13.0 |
| | | UTDC* | 6.6 | 6.8 | 4.8 | 6.5 | 5.9 | 4.8 | 7.0 | 7.8 | 7.4 | 8.9 | 6.9 | 4.3 | 6.5 |
| | | Target-TS | 5.6 | 5.5 | 3.3 | 4.3 | 5.0 | 3.9 | 5.4 | 7.0 | 3.0 | 4.6 | 6.5 | 3.1 | 4.8 |
| | | Uncalibrated | 27.7 | 12.2 | 7.5 | 10.3 | 11.8 | 10.4 | 9.3 | 29.9 | 6.5 | 7.2 | 27.0 | 7.1 | 13.9 |
| | | Source-TS | 32.8 | 15.0 | 11.2 | 16.6 | 14.3 | 14.0 | 14.3 | 35.0 | 10.6 | 11.7 | 30.8 | 10.3 | 18.0 |
| | | PseudoCal | 11.6 | 7.9 | 4.2 | 6.9 | 6.3 | 5.2 | 7.5 | 15.8 | 4.3 | 5.6 | 13.6 | 4.4 | 7.8 |
| | | PLCC* | 6.3 | 7.5 | 4.3 | 9.8 | 7.9 | 6.3 | 8.9 | 7.4 | 4.0 | 7.6 | 7.4 | 3.8 | **6.6** |
| | | PLCC | 8.3 | 6.8 | 4.2 | 6.4 | 6.2 | 5.3 | 6.5 | 12.3 | 3.8 | 6.2 | 10.1 | 3.7 | **6.6** |
| AaD | Oracle | CPCS | 28.1 | 10.2 | 10.4 | 22.3 | 13.0 | 7.0 | 20.1 | 23.9 | 11.1 | 11.5 | 19.9 | 6.3 | 15.3 |
| | | TransCal | 24.7 | 8.2 | 10.6 | 15.3 | 9.0 | 6.8 | 12.9 | 16.6 | 10.8 | 10.7 | 13.4 | 5.4 | 12.0 |
| | | UTDC* | 9.1 | 7.7 | 5.2 | 5.9 | 7.5 | 5.9 | 8.5 | 10.6 | 5.6 | 7.6 | 9.9 | 5.4 | 7.4 |
| | | Target-TS | 8.3 | 6.5 | 4.5 | 4.7 | 6.5 | 4.8 | 7.8 | 9.6 | 4.3 | 5.0 | 9.1 | 4.4 | 6.3 |
| | | Uncalibrated | 31.4 | 13.7 | 11.6 | 14.0 | 14.9 | 13.3 | 15.5 | 30.8 | 10.8 | 11.5 | 29.6 | 9.4 | 17.2 |
| | | Source-TS | 33.9 | 15.8 | 14.0 | 18.6 | 17.4 | 15.9 | 19.4 | 34.4 | 14.2 | 13.3 | 32.5 | 10.7 | 20.0 |
| | | PseudoCal | 15.6 | 9.4 | 5.5 | 6.9 | 8.5 | 6.4 | 10.5 | 16.8 | 5.2 | 6.6 | 14.2 | 5.2 | 9.2 |
| | | PLCC* | 10.5 | 9.9 | 5.0 | 7.1 | 11.8 | 9.0 | 12.5 | 13.5 | 4.8 | 6.1 | 12.4 | 5.9 | 9.0 |
| | | PLCC | 10.3 | 6.8 | 5.8 | 5.7 | 7.1 | 5.8 | 8.3 | 11.1 | 5.5 | 5.9 | 11.1 | 4.5 | **7.3** |

Table 7: ECE for top-1 predictions (in %) on **VisDA**, using 15 bins (with the lowest in bold) across various SFDA methods with different calibration methods.

| Type | Method | DCPL | SHOT | AaD |
|------|--------|------|------|-----|
| Oracle | CPCS | 12.3 | 19.2 | 11.6 |
| | TransCal | 10.5 | 14.4 | 9.2 |
| | UTDC* | 4.6 | 3.2 | 3.2 |
| | Target-TS | 4.0 | 2.9 | 2.8 |
| | Uncalibrated | 13.7 | 15.5 | 12.1 |
| | Source-TS | 15.3 | 18.1 | 13.2 |
| | NTS | 12.3 | N/A | N/A |
| | PseudoCal | 9.5 | 5.6 | 10.2 |
| | PLCC* | 29.4 | 26.1 | 27.8 |
| | PLCC | **5.6** | **4.2** | **4.8** |

Table 8: ECE for top-1 predictions (in %) on **DomainNet40**, using 15 bins (with the lowest in bold) across various SFDA classification tasks and methods with different calibration methods.

| SFDA | Type | Method | CP | CR | CS | PC | PR | PS | RC | RP | RS | SC | SP | SR | Avg |
|---|---|---|---|---|---|---|---|---|---|---|---|---|---|---|---|
| DCPL | Oracle | CPCS | 14.1 | 1.7 | 11.9 | 14.7 | 5.9 | 10.2 | 10.1 | 5.6 | 14.8 | 9.3 | 10.6 | 13.6 | 10.2 |
| | | TransCal | 5.5 | 8.2 | 6.8 | 5.8 | 11.9 | 9.6 | 4.8 | 4.4 | 11.0 | 8.0 | 9.0 | 12.9 | 8.2 |
| | | UTDC* | 3.1 | 12.6 | 4.4 | 4.7 | 3.6 | 3.5 | 3.9 | 4.0 | 4.6 | 5.2 | 2.9 | 4.6 | 4.8 |
| | | Target-TS | 2.3 | 0.7 | 3.8 | 3.9 | 0.9 | 2.8 | 3.3 | 1.9 | 3.0 | 4.5 | 2.5 | 1.8 | 2.6 |
| | | Uncalibrated | 5.9 | 2.4 | 8.6 | 5.5 | 3.6 | 9.8 | 4.4 | 6.2 | 12.4 | 6.1 | 7.2 | 4.5 | 6.4 |
| | | Source-TS | 12.5 | 5.0 | 15.1 | 11.9 | 5.8 | 17.0 | 11.1 | 11.7 | 19.5 | 8.0 | 13.1 | 6.8 | 11.5 |
| | | NTS | 14.5 | 5.5 | 16.1 | 17.8 | 5.4 | 15.4 | 14.8 | 12.0 | 16.4 | 14.7 | 14.8 | 7.0 | 12.9 |
| | | PseudoCal | 3.0 | 6.3 | 4.3 | 5.3 | 5.8 | 3.6 | 6.1 | 5.1 | 4.9 | 5.9 | 4.3 | 5.0 | 5.0 |
| | | PLCC* | 19.6 | 13.2 | 12.3 | 10.2 | 5.6 | 6.3 | 7.0 | 2.3 | 7.9 | 11.2 | 9.8 | 10.4 | 9.6 |
| | | PLCC | 2.7 | 0.8 | 5.3 | 4.3 | 2.1 | 4.1 | 3.8 | 2.4 | 3.8 | 4.7 | 2.9 | 2.7 | **3.3** |
| SHOT | Oracle | CPCS | 13.5 | 8.0 | 12.8 | 20.8 | 3.7 | 6.5 | 6.5 | 4.3 | 9.3 | 10.7 | 16.3 | 16.9 | 10.8 |
| | | TransCal | 6.0 | 12.9 | 4.1 | 5.9 | 5.3 | 10.0 | 5.2 | 2.1 | 4.6 | 5.8 | 4.5 | 16.1 | 6.9 |
| | | UTDC* | 4.2 | 7.3 | 6.5 | 5.0 | 2.5 | 3.4 | 8.8 | 7.8 | 9.1 | 4.6 | 2.6 | 3.8 | 5.5 |
| | | Target-TS | 2.9 | 1.2 | 3.0 | 4.1 | 1.9 | 2.2 | 3.8 | 1.7 | 2.4 | 4.1 | 2.4 | 1.9 | 2.6 |
| | | Uncalibrated | 3.1 | 1.5 | 4.1 | 4.2 | 3.3 | 4.0 | 5.2 | 2.1 | 4.6 | 4.4 | 3.9 | 3.0 | 3.6 |
| | | Source-TS | 11.0 | 5.2 | 12.3 | 11.1 | 6.8 | 13.5 | 8.1 | 9.2 | 15.1 | 6.4 | 11.7 | 6.6 | 9.8 |
| | | PseudoCal | 3.4 | 3.7 | 3.7 | 4.8 | 2.8 | 3.5 | 6.4 | 2.9 | 4.9 | 5.0 | 3.0 | 3.6 | 4.0 |
| | | PLCC* | 16.1 | 11.6 | 9.0 | 6.6 | 2.5 | 5.1 | 5.2 | 3.9 | 6.0 | 7.9 | 8.1 | 8.5 | 7.5 |
| | | PLCC | 3.0 | 1.4 | 3.9 | 4.4 | 2.2 | 4.5 | 4.5 | 2.0 | 4.8 | 4.4 | 2.9 | 2.5 | **3.4** |
| AaD | Oracle | CPCS | 12.1 | 1.6 | 14.3 | 12.7 | 3.9 | 4.1 | 6.8 | 7.5 | 9.0 | 12.9 | 13.0 | 23.7 | 10.1 |
| | | TransCal | 10.8 | 11.1 | 4.6 | 14.9 | 4.9 | 7.7 | 4.6 | 4.1 | 6.7 | 6.1 | 8.3 | 17.7 | 8.5 |
| | | UTDC* | 2.4 | 8.3 | 3.4 | 4.7 | 3.0 | 2.4 | 9.1 | 6.9 | 6.9 | 4.2 | 2.9 | 6.4 | 5.0 |
| | | Target-TS | 2.1 | 1.0 | 2.6 | 3.9 | 0.6 | 1.8 | 3.3 | 2.2 | 2.2 | 3.8 | 2.8 | 0.9 | 2.3 |
| | | Uncalibrated | 3.3 | 2.8 | 4.6 | 4.1 | 3.4 | 5.5 | 3.7 | 4.1 | 6.7 | 4.1 | 4.2 | 2.4 | **4.1** |
| | | Source-TS | 11.1 | 5.5 | 12.0 | 10.0 | 5.7 | 13.8 | 9.2 | 10.6 | 15.6 | 5.7 | 10.3 | 4.7 | 9.5 |
| | | PseudoCal | 2.9 | 3.5 | 3.8 | 4.7 | 3.8 | 3.2 | 5.6 | 3.7 | 3.8 | 6.0 | 3.1 | 5.8 | 4.2 |
| | | PLCC* | 16.9 | 12.7 | 12.2 | 9.3 | 3.8 | 5.4 | 5.5 | 3.0 | 8.3 | 13.0 | 12.6 | 13.4 | 9.7 |
| | | PLCC | 4.1 | 1.1 | 6.6 | 5.2 | 1.1 | 7.1 | 6.1 | 3.7 | 6.2 | 6.2 | 5.0 | 2.0 | 4.5 |

Table 9: ECE for top-1 predictions (in %) on **DomainNet126**, using 15 bins (with the lowest in bold) across various SFDA classification tasks and methods with different calibration methods.

| SFDA | Type | Method | CR | CS | PC | PR | PS | RC | RS | SC | SR | Avg |
|---|---|---|---|---|---|---|---|---|---|---|---|---|
| DCPL | Oracle | CPCS | 11.6 | 20.5 | 14.2 | 8.8 | 22.5 | 14.3 | 16.1 | 10.0 | 8.1 | 14.0 |
| | | TransCal | 6.7 | 20.1 | 11.2 | 5.6 | 10.2 | 13.9 | 19.0 | 10.1 | 7.2 | 11.6 |
| | | UTDC* | 6.7 | 5.9 | 6.6 | 4.2 | 5.6 | 5.5 | 6.8 | 4.8 | 6.8 | 5.9 |
| | | Target-TS | 4.3 | 5.1 | 6.1 | 3.6 | 5.4 | 4.8 | 6.2 | 4.4 | 4.0 | 4.9 |
| | | Uncalibrated | 13.1 | 24.5 | 19.1 | 11.6 | 23.9 | 16.4 | 26.5 | 13.7 | 13.8 | 18.1 |
| | | Source-TS | 15.4 | 29.5 | 25.0 | 14.0 | 29.1 | 21.0 | 30.9 | 18.9 | 15.9 | 22.2 |
| | | NTS | 14.7 | 20.5 | 20.3 | 13.4 | 18.3 | 16.4 | 22.6 | 13.6 | 14.5 | 17.1 |
| | | PseudoCal | 4.5 | 6.5 | 7.7 | 3.9 | 5.8 | 6.3 | 6.8 | 5.0 | 4.4 | **5.7** |
| | | PLCC* | 11.2 | 6.9 | 11.8 | 6.1 | 14.0 | 7.1 | 8.5 | 7.4 | 11.4 | 9.4 |
| | | PLCC | 5.8 | 5.4 | 6.3 | 5.1 | 5.9 | 5.6 | 6.6 | 5.2 | 6.0 | 5.8 |
| SHOT | Oracle | CPCS | 12.5 | 19.3 | 21.7 | 11.8 | 23.4 | 17.6 | 16.2 | 12.1 | 6.8 | 15.7 |
| | | TransCal | 11.6 | 16.7 | 15.5 | 10.3 | 12.5 | 13.4 | 20.8 | 8.6 | 4.5 | 12.7 |
| | | UTDC* | 4.5 | 5.3 | 7.6 | 3.9 | 5.2 | 6.9 | 5.4 | 8.1 | 3.7 | 5.6 |
| | | Target-TS | 4.4 | 4.8 | 6.3 | 3.4 | 4.2 | 5.8 | 5.1 | 4.8 | 3.4 | 4.7 |
| | | Uncalibrated | 12.7 | 16.7 | 15.5 | 10.8 | 17.9 | 13.4 | 22.2 | 8.6 | 13.4 | 14.6 |
| | | Source-TS | 16.7 | 24.9 | 24.5 | 14.8 | 26.5 | 20.8 | 29.4 | 16.1 | 17.4 | 21.2 |
| | | PseudoCal | 5.4 | 6.3 | 9.6 | 4.3 | 5.6 | 6.3 | 5.6 | 5.2 | 4.6 | 5.9 |
| | | PLCC* | 7.9 | 5.2 | 8.9 | 3.8 | 9.9 | 6.3 | 6.0 | 5.5 | 7.5 | 6.8 |
| | | PLCC | 5.2 | 5.0 | 6.7 | 4.8 | 4.4 | 6.4 | 5.2 | 5.2 | 4.8 | **5.3** |
| AaD | Oracle | CPCS | 15.4 | 17.2 | 13.4 | 10.2 | 15.5 | 12.5 | 13.0 | 8.8 | 7.7 | 12.6 |
| | | TransCal | 10.2 | 20.9 | 14.2 | 7.6 | 7.6 | 11.2 | 14.3 | 9.3 | 7.9 | 11.5 |
| | | UTDC* | 7.6 | 5.7 | 7.8 | 5.3 | 5.2 | 6.7 | 6.6 | 4.4 | 7.0 | 6.3 |
| | | Target-TS | 6.7 | 5.0 | 7.1 | 4.7 | 5.1 | 5.3 | 5.3 | 3.6 | 5.8 | 5.4 |
| | | Uncalibrated | 17.0 | 22.9 | 21.5 | 12.3 | 22.4 | 15.9 | 26.5 | 10.0 | 15.8 | 18.3 |
| | | Source-TS | 19.5 | 29.1 | 28.4 | 15.0 | 29.0 | 21.8 | 32.1 | 16.2 | 18.4 | 23.3 |
| | | PseudoCal | 8.0 | 8.8 | 11.8 | 4.9 | 7.3 | 5.8 | 5.8 | 4.3 | 6.3 | **7.0** |
| | | PLCC* | 9.3 | 6.5 | 10.9 | 6.1 | 12.0 | 8.3 | 7.9 | 7.3 | 9.7 | 8.7 |
| | | PLCC | 6.9 | 6.3 | 7.4 | 5.2 | 7.1 | 8.0 | 7.8 | 7.9 | 6.0 | **7.0** |

### A.1.3 NEGATIVE LOG-LIKELIHOOD RESULTS

The Negative Log-Likelihood (NLL) (Hastie et al., 2009) is a loss function commonly used in probabilistic models to assess how accurately a probabilistic distribution predicts a set of outcomes. A lower NLL value indicates better model performance, as it minimizes the negative logarithm of the predicted probabilities for the observed data. Tables 10, 11, 12, and 13 provide the NLL calibration results for Office-Home, VisDA, DomainNet40, and DomainNet126, respectively.

Table 10: 100 * NLL on **Office-Home**, using 15 bins (with the lowest in bold) across various SFDA classification tasks and methods with different calibration methods.

| SFDA | Type | Method | AC | AP | AR | CA | CP | CR | PA | PC | PR | RA | RC | RP | Avg |
|------|------|--------|----|----|----|----|----|----|----|----|----|----|----|----|-----|
| DCPL | Oracle | CPCS | 321 | 166 | 104 | 453 | 92 | 76 | 697 | 240 | 201 | 132 | 193 | 65 | 228 |
| | | TransCal | 176 | 90 | 65 | 132 | 82 | 74 | 133 | 200 | 79 | 115 | 181 | 60 | 116 |
| | | UTDC* | 165 | 87 | 64 | 120 | 82 | 75 | 130 | 183 | 77 | 120 | 167 | 60 | 111 |
| | | Target-TS | 164 | 87 | 65 | 120 | 82 | 74 | 124 | 183 | 74 | 109 | 167 | 60 | 109 |
| | | Uncalibrated | 222 | 104 | 68 | 128 | 97 | 80 | 133 | 256 | 79 | 115 | 228 | 66 | 131 |
| | | Source-TS | 305 | 141 | 86 | 160 | 129 | 102 | 163 | 346 | 100 | 142 | 309 | 87 | 172 |
| | | NTS | 224 | 101 | 69 | 148 | 97 | 77 | 131 | 238 | 86 | 126 | 214 | 66 | 131 |
| | | PseudoCal | 167 | 87 | 66 | 120 | 82 | 74 | 124 | 186 | 74 | 109 | 168 | 60 | **110** |
| | | PLCC* | 171 | 92 | 69 | 134 | 95 | 85 | 137 | 188 | 75 | 109 | 172 | 64 | 116 |
| | | PLCC | 164 | 87 | 64 | 120 | 83 | 74 | 125 | 185 | 74 | 110 | 169 | 60 | **110** |
| SHOT | Oracle | CPCS | 509 | 202 | 160 | 201 | 388 | 102 | 433 | 388 | 172 | 173 | 231 | 179 | 262 |
| | | TransCal | 246 | 117 | 84 | 156 | 114 | 101 | 150 | 270 | 85 | 124 | 207 | 86 | 145 |
| | | UTDC* | 201 | 112 | 81 | 139 | 108 | 99 | 145 | 218 | 86 | 132 | 195 | 79 | 133 |
| | | Target-TS | 200 | 112 | 82 | 140 | 108 | 98 | 145 | 217 | 83 | 122 | 195 | 79 | 132 |
| | | Uncalibrated | 256 | 130 | 84 | 145 | 124 | 105 | 150 | 281 | 85 | 124 | 252 | 87 | 152 |
| | | Source-TS | 346 | 173 | 104 | 178 | 165 | 133 | 180 | 375 | 104 | 149 | 339 | 113 | 196 |
| | | PseudoCal | 206 | 113 | 81 | 139 | 108 | 98 | 145 | 221 | 83 | 121 | 198 | 79 | 133 |
| | | PLCC* | 201 | 112 | 81 | 144 | 111 | 101 | 148 | 217 | 83 | 121 | 195 | 79 | 133 |
| | | PLCC | 200 | 112 | 81 | 140 | 108 | 98 | 145 | 218 | 82 | 123 | 195 | 79 | **132** |
| AaD | Oracle | CPCS | 292 | 126 | 109 | 232 | 195 | 111 | 605 | 276 | 118 | 228 | 661 | 92 | 254 |
| | | TransCal | 253 | 119 | 107 | 174 | 128 | 111 | 196 | 229 | 113 | 152 | 213 | 90 | 157 |
| | | UTDC* | 214 | 118 | 96 | 159 | 124 | 110 | 176 | 219 | 97 | 142 | 207 | 89 | 146 |
| | | Target-TS | 214 | 118 | 96 | 159 | 124 | 110 | 176 | 219 | 97 | 138 | 207 | 89 | 146 |
| | | Uncalibrated | 335 | 159 | 114 | 189 | 169 | 133 | 219 | 343 | 113 | 158 | 318 | 110 | 197 |
| | | Source-TS | 466 | 219 | 153 | 247 | 232 | 177 | 286 | 468 | 148 | 203 | 437 | 148 | 265 |
| | | PseudoCal | 221 | 119 | 96 | 159 | 127 | 110 | 179 | 228 | 97 | 139 | 212 | 89 | 148 |
| | | PLCC* | 215 | 118 | 97 | 161 | 127 | 114 | 180 | 222 | 97 | 139 | 210 | 89 | 147 |
| | | PLCC | 214 | 119 | 96 | 159 | 126 | 110 | 175 | 220 | 97 | 138 | 209 | 89 | **146** |

Table 11: 100 * NLL on **VisDA**, using 15 bins (with the lowest in bold) across various SFDA methods with different calibration methods.

| Type | Method | DCPL | SHOT | AaD |
|------|--------|------|------|-----|
| Oracle | CPCS | 90 | 572 | 84 |
| | TransCal | 94 | 106 | 81 |
| | UTDC* | 73 | 84 | 62 |
| | Target-TS | 73 | 84 | 62 |
| | Uncalibrated | 107 | 114 | 113 |
| | Source-TS | 162 | 171 | 173 |
| | NTS | 75 | N/A | N/A |
| | PseudoCal | 76 | **85** | 66 |
| | PLCC* | 97 | 104 | 85 |
| | PLCC | **74** | **85** | **62** |

Table 12: 100 * NLL on **DomainNet40**, using 15 bins (with the lowest in bold) across various SFDA classification tasks and methods with different calibration methods.

| SFDA | Type | Method | CP | CR | CS | PC | PR | PS | RC | RP | RS | SC | SP | SR | Avg |
|---|---|---|---|---|---|---|---|---|---|---|---|---|---|---|---|
| DCPL | Oracle | CPCS | 295 | 34 | 179 | 173 | 44 | 205 | 105 | 82 | 186 | 100 | 253 | 64 | 143 |
| | | TransCal | 84 | 39 | 103 | 92 | 50 | 115 | 88 | 79 | 126 | 92 | 93 | 60 | 85 |
| | | UTDC* | 81 | 44 | 102 | 91 | 42 | 110 | 88 | 79 | 120 | 90 | 88 | 52 | 82 |
| | | Target-TS | 81 | 34 | 101 | 91 | 41 | 110 | 88 | 78 | 120 | 90 | 88 | 51 | 81 |
| | | Uncalibrated | 83 | 34 | 106 | 91 | 43 | 116 | 88 | 80 | 128 | 91 | 92 | 55 | 84 |
| | | Source-TS | 111 | 45 | 136 | 105 | 52 | 155 | 104 | 100 | 170 | 97 | 125 | 70 | 106 |
| | | NTS | 327 | 53 | 273 | 421 | 53 | 140 | 222 | 110 | 143 | 290 | 450 | 73 | 213 |
| | | PseudoCal | 81 | 37 | 102 | 93 | 44 | 110 | 90 | 80 | 120 | 91 | 89 | 52 | 82 |
| | | PLCC* | 97 | 44 | 108 | 98 | 44 | 112 | 91 | 78 | 122 | 95 | 92 | 56 | 87 |
| | | PLCC | 81 | 34 | 102 | 91 | 41 | 111 | 88 | 78 | 120 | 90 | 88 | 52 | **81** |
| SHOT | Oracle | CPCS | 186 | 53 | 139 | 476 | 54 | 110 | 95 | 81 | 124 | 207 | 343 | 97 | 164 |
| | | TransCal | 93 | 50 | 105 | 112 | 56 | 110 | 94 | 80 | 117 | 97 | 90 | 72 | 90 |
| | | UTDC* | 86 | 44 | 106 | 107 | 53 | 108 | 101 | 89 | 121 | 96 | 89 | 55 | 88 |
| | | Target-TS | 86 | 39 | 105 | 108 | 53 | 108 | 93 | 80 | 117 | 95 | 89 | 54 | 86 |
| | | Uncalibrated | 86 | 39 | 105 | 108 | 53 | 108 | 94 | 80 | 117 | 95 | 89 | 54 | **86** |
| | | Source-TS | 102 | 46 | 120 | 115 | 62 | 128 | 99 | 89 | 138 | 98 | 110 | 65 | 98 |
| | | PseudoCal | 86 | 41 | 105 | 108 | 54 | 108 | 95 | 81 | 118 | 96 | 89 | 55 | **86** |
| | | PLCC* | 99 | 49 | 110 | 110 | 53 | 110 | 94 | 80 | 119 | 99 | 93 | 58 | 90 |
| | | PLCC | 86 | 39 | 106 | 108 | 53 | 109 | 94 | 80 | 118 | 96 | 89 | 54 | **86** |
| AaD | Oracle | CPCS | 234 | 37 | 282 | 138 | 42 | 102 | 95 | 87 | 117 | 420 | 212 | 70 | 153 |
| | | TransCal | 90 | 47 | 98 | 111 | 42 | 106 | 92 | 80 | 113 | 88 | 85 | 56 | 84 |
| | | UTDC* | 80 | 43 | 98 | 95 | 41 | 101 | 101 | 86 | 113 | 85 | 81 | 40 | 80 |
| | | Target-TS | 80 | 37 | 97 | 95 | 40 | 101 | 92 | 80 | 111 | 85 | 81 | 36 | 78 |
| | | Uncalibrated | 80 | 38 | 98 | 95 | 41 | 103 | 92 | 80 | 113 | 85 | 82 | 37 | **79** |
| | | Source-TS | 99 | 49 | 117 | 102 | 50 | 128 | 100 | 94 | 138 | 88 | 103 | 44 | 93 |
| | | PseudoCal | 80 | 38 | 98 | 96 | 41 | 102 | 93 | 81 | 112 | 86 | 81 | 39 | **79** |
| | | PLCC* | 95 | 48 | 105 | 101 | 42 | 104 | 94 | 80 | 116 | 93 | 89 | 47 | 84 |
| | | PLCC | 81 | 37 | 100 | 96 | 40 | 105 | 94 | 81 | 114 | 87 | 82 | 37 | 80 |

Table 13: 100 * NLL on **DomainNet126**, using 15 bins (with the lowest in bold) across various SFDA classification tasks and methods with different calibration methods.

| SFDA | Type | Method | CR | CS | PC | PR | PS | RC | RS | SC | SR | Avg |
|------|------|--------|----|----|----|----|----|----|----|----|----|-----|
| DCPL | Oracle | CPCS | 122 | 225 | 180 | 106 | 210 | 156 | 215 | 143 | 114 | 163 |
| | | TransCal | 107 | 223 | 173 | 98 | 195 | 155 | 222 | 140 | 110 | 158 |
| | | UTDC* | 106 | 197 | 169 | 96 | 192 | 144 | 205 | 134 | 109 | 150 |
| | | Target-TS | 105 | 197 | 169 | 96 | 192 | 144 | 205 | 134 | 108 | 150 |
| | | Uncalibrated | 132 | 248 | 196 | 120 | 240 | 164 | 263 | 150 | 139 | 183 |
| | | Source-TS | 176 | 326 | 253 | 160 | 316 | 206 | 340 | 190 | 185 | 239 |
| | | NTS | 158 | 224 | 203 | 145 | 211 | 164 | 238 | 150 | 149 | 182 |
| | | PseudoCal | 105 | 198 | 169 | 96 | 192 | 145 | 205 | 134 | 108 | **150** |
| | | PLCC* | 109 | 198 | 175 | 97 | 199 | 146 | 206 | 136 | 112 | 153 |
| | | PLCC | 106 | 197 | 169 | 97 | 192 | 145 | 205 | 135 | 109 | 151 |
| SHOT | Oracle | CPCS | 132 | 216 | 232 | 124 | 693 | 178 | 218 | 151 | 119 | 229 |
| | | TransCal | 129 | 205 | 197 | 114 | 201 | 166 | 230 | 142 | 118 | 167 |
| | | UTDC* | 116 | 189 | 186 | 104 | 194 | 158 | 208 | 142 | 118 | 157 |
| | | Target-TS | 117 | 189 | 186 | 104 | 194 | 158 | 208 | 139 | 118 | 157 |
| | | Uncalibrated | 132 | 205 | 197 | 116 | 211 | 166 | 235 | 142 | 135 | 171 |
| | | Source-TS | 171 | 254 | 241 | 150 | 264 | 199 | 293 | 169 | 174 | 213 |
| | | PseudoCal | 117 | 189 | 187 | 104 | 193 | 158 | 208 | 139 | 117 | **157** |
| | | PLCC* | 120 | 190 | 190 | 104 | 199 | 159 | 209 | 140 | 121 | 159 |
| | | PLCC | 117 | 190 | 186 | 104 | 194 | 159 | 208 | 139 | 118 | **157** |
| AaD | Oracle | CPCS | 167 | 223 | 204 | 121 | 228 | 166 | 225 | 140 | 131 | 178 |
| | | TransCal | 144 | 236 | 208 | 111 | 202 | 164 | 228 | 139 | 129 | 173 |
| | | UTDC* | 136 | 203 | 198 | 106 | 201 | 159 | 220 | 133 | 128 | 165 |
| | | Target-TS | 137 | 203 | 198 | 106 | 201 | 159 | 220 | 133 | 129 | 165 |
| | | Uncalibrated | 184 | 246 | 230 | 134 | 242 | 175 | 272 | 140 | 170 | 199 |
| | | Source-TS | 245 | 319 | 295 | 179 | 314 | 217 | 347 | 171 | 226 | 257 |
| | | NLL | 140 | 205 | 202 | 106 | 202 | 159 | 220 | 133 | 130 | **166** |
| | | PLCC* | 137 | 204 | 203 | 106 | 207 | 162 | 221 | 135 | 130 | 167 |
| | | PLCC | 138 | 204 | 198 | 107 | 202 | 161 | 221 | 136 | 130 | **166** |

## A.1.4    BRIER SCORE RESULTS

The Brier Score (BS) (Brier, 1950) is a metric that evaluates the accuracy of probabilistic predictions by calculating the mean squared difference between the predicted probabilities and the true label $y$, where lower scores signify better predictive performance. Tables 14, 15, 16, and 17 provide the BS calibration results for Office-Home, VisDA, DomainNet40, and DomainNet126, respectively.

Table 14: 100 * BS on **Office-Home**, using 15 bins (with the lowest in bold) across various SFDA classification tasks and methods with different calibration methods.

| SFDA | Type | Method | AC | AP | AR | CA | CP | CR | PA | PC | PR | RA | RC | RP | Avg |
|------|------|--------|----|----|----|----|----|----|----|----|----|----|----|----|-----|
| DCPL | Oracle | CPCS | 56 | 29 | 25 | 49 | 26 | 26 | 50 | 62 | 28 | 39 | 54 | 20 | 39 |
| | | TransCal | 53 | 28 | 24 | 43 | 26 | 26 | 42 | 59 | 26 | 37 | 53 | 20 | 36 |
| | | UTDC* | 51 | 28 | 23 | 40 | 26 | 26 | 41 | 56 | 26 | 38 | 52 | 20 | 36 |
| | | Target-TS | 51 | 28 | 24 | 40 | 26 | 26 | 41 | 56 | 26 | 37 | 51 | 20 | 36 |
| | | Uncalibrated | 56 | 28 | 24 | 41 | 26 | 27 | 42 | 64 | 26 | 37 | 57 | 20 | 37 |
| | | Source-TS | 60 | 29 | 25 | 44 | 27 | 28 | 44 | 67 | 28 | 40 | 60 | 21 | 39 |
| | | NTS | 56 | 28 | 24 | 43 | 26 | 26 | 41 | 62 | 27 | 38 | 56 | 20 | 37 |
| | | PseudoCal | 51 | 28 | 24 | 40 | 26 | 26 | 41 | 57 | 26 | 37 | 51 | 20 | 36 |
| | | PLCC* | 53 | 30 | 25 | 44 | 30 | 29 | 46 | 58 | 26 | 37 | 53 | 21 | 38 |
| | | PLCC | 51 | 28 | 23 | 40 | 26 | 26 | 41 | 57 | 26 | 37 | 52 | 19 | **35** |
| SHOT | Oracle | CPCS | 75 | 40 | 35 | 53 | 40 | 36 | 58 | 80 | 33 | 45 | 65 | 29 | 49 |
| | | TransCal | 71 | 37 | 31 | 53 | 36 | 36 | 49 | 75 | 30 | 42 | 64 | 27 | 46 |
| | | UTDC* | 63 | 37 | 30 | 48 | 36 | 36 | 48 | 67 | 30 | 42 | 61 | 26 | 44 |
| | | Target-TS | 63 | 37 | 30 | 48 | 36 | 35 | 48 | 66 | 30 | 41 | 61 | 26 | 44 |
| | | Uncalibrated | 72 | 38 | 31 | 49 | 38 | 36 | 49 | 77 | 30 | 42 | 69 | 27 | 46 |
| | | Source-TS | 77 | 40 | 33 | 53 | 40 | 39 | 52 | 81 | 32 | 44 | 74 | 28 | 49 |
| | | PseudoCal | 65 | 37 | 30 | 48 | 36 | 35 | 48 | 68 | 30 | 41 | 62 | 26 | **44** |
| | | PLCC* | 63 | 37 | 30 | 49 | 37 | 36 | 49 | 66 | 30 | 41 | 61 | 26 | **44** |
| | | PLCC | 63 | 37 | 30 | 48 | 36 | 36 | 48 | 67 | 30 | 41 | 61 | 26 | **44** |
| AaD | Oracle | CPCS | 74 | 38 | 36 | 60 | 41 | 38 | 64 | 72 | 35 | 49 | 71 | 28 | 50 |
| | | TransCal | 71 | 37 | 35 | 55 | 39 | 38 | 58 | 68 | 35 | 47 | 64 | 28 | 48 |
| | | UTDC* | 65 | 37 | 34 | 51 | 39 | 37 | 56 | 66 | 33 | 46 | 63 | 28 | 46 |
| | | Target-TS | 65 | 37 | 34 | 51 | 39 | 37 | 56 | 66 | 33 | 46 | 63 | 28 | 46 |
| | | Uncalibrated | 76 | 39 | 36 | 55 | 42 | 40 | 60 | 77 | 35 | 47 | 73 | 29 | 51 |
| | | Source-TS | 80 | 40 | 38 | 59 | 44 | 42 | 63 | 81 | 37 | 51 | 77 | 30 | 54 |
| | | PseudoCal | 67 | 37 | 34 | 51 | 39 | 37 | 56 | 68 | 33 | 46 | 64 | 28 | 47 |
| | | PLCC* | 65 | 38 | 34 | 52 | 40 | 38 | 57 | 67 | 33 | 46 | 64 | 29 | 47 |
| | | PLCC | 65 | 37 | 34 | 51 | 39 | 37 | 56 | 66 | 33 | 46 | 64 | 28 | **46** |

Table 15: 100 * BS on **VisDA**, using 15 bins (with the lowest in bold) across various SFDA methods with different calibration methods.

| Type | Method | DCPL | SHOT | AaD |
|------|--------|------|------|-----|
| Oracle | CPCS | 30 | 40 | 26 |
| | TransCal | 30 | 36 | 25 |
| | UTDC* | 28 | 33 | 24 |
| | Target-TS | 28 | 33 | 24 |
| | Uncalibrated | 31 | 36 | 27 |
| | Source-TS | 32 | 39 | 28 |
| | NTS | **28** | N/A | N/A |
| | PseudoCal | 29 | **33** | 25 |
| | PLCC* | 39 | 41 | 34 |
| | PLCC | **28** | **33** | **24** |

Table 16: 100 * BS on **DomainNet40**, using 15 bins (with the lowest in bold) across various SFDA classification tasks and methods with different calibration methods.

| SFDA | Type | Method | CP | CR | CS | PC | PR | PS | RC | RP | RS | SC | SP | SR | Avg |
|---|---|---|---|---|---|---|---|---|---|---|---|---|---|---|---|
| DCPL | Oracle | CPCS | 33 | 12 | 38 | 39 | 15 | 39 | 32 | 27 | 44 | 32 | 32 | 22 | 30 |
| | | TransCal | 30 | 13 | 35 | 34 | 16 | 38 | 30 | 27 | 42 | 32 | 31 | 19 | 29 |
| | | UTDC* | 28 | 14 | 35 | 34 | 14 | 37 | 30 | 27 | 40 | 31 | 29 | 16 | 28 |
| | | Target-TS | 28 | 12 | 35 | 34 | 14 | 37 | 30 | 27 | 40 | 31 | 29 | 16 | 28 |
| | | Uncalibrated | 29 | 12 | 36 | 34 | 14 | 38 | 30 | 27 | 42 | 31 | 30 | 16 | **28** |
| | | Source-TS | 31 | 13 | 38 | 36 | 15 | 41 | 32 | 29 | 46 | 32 | 32 | 17 | 30 |
| | | NTS | 33 | 13 | 40 | 41 | 15 | 40 | 36 | 30 | 44 | 36 | 34 | 17 | 32 |
| | | PseudoCal | 28 | 13 | 35 | 34 | 15 | 37 | 31 | 27 | 40 | 31 | 30 | 17 | **28** |
| | | PLCC* | 33 | 14 | 37 | 36 | 15 | 37 | 31 | 27 | 41 | 33 | 31 | 18 | 29 |
| | | PLCC | 28 | 12 | 35 | 34 | 14 | 37 | 30 | 27 | 40 | 31 | 30 | 16 | **28** |
| SHOT | Oracle | CPCS | 34 | 16 | 40 | 50 | 19 | 37 | 34 | 28 | 40 | 36 | 36 | 25 | 33 |
| | | TransCal | 32 | 17 | 37 | 40 | 19 | 38 | 33 | 28 | 39 | 34 | 30 | 24 | 31 |
| | | UTDC* | 31 | 16 | 37 | 40 | 19 | 37 | 33 | 29 | 40 | 33 | 30 | 18 | 30 |
| | | Target-TS | 31 | 15 | 37 | 40 | 19 | 37 | 32 | 28 | 39 | 33 | 30 | 18 | 30 |
| | | Uncalibrated | 31 | 15 | 37 | 40 | 19 | 37 | 33 | 28 | 39 | 33 | 30 | 18 | **30** |
| | | Source-TS | 33 | 16 | 39 | 41 | 19 | 40 | 33 | 29 | 42 | 33 | 32 | 18 | 31 |
| | | PseudoCal | 31 | 15 | 37 | 40 | 19 | 37 | 33 | 28 | 39 | 33 | 30 | 18 | **30** |
| | | PLCC* | 34 | 17 | 38 | 40 | 19 | 37 | 33 | 28 | 40 | 34 | 31 | 19 | 31 |
| | | PLCC | 31 | 15 | 37 | 40 | 19 | 37 | 33 | 28 | 39 | 33 | 30 | 18 | **30** |
| AaD | Oracle | CPCS | 33 | 14 | 40 | 39 | 15 | 35 | 33 | 29 | 40 | 36 | 32 | 25 | 31 |
| | | TransCal | 31 | 16 | 34 | 39 | 15 | 36 | 32 | 28 | 38 | 31 | 29 | 20 | 29 |
| | | UTDC* | 29 | 15 | 34 | 35 | 15 | 35 | 33 | 29 | 38 | 30 | 28 | 14 | 28 |
| | | Target-TS | 29 | 14 | 34 | 35 | 14 | 35 | 32 | 28 | 38 | 30 | 28 | 13 | 28 |
| | | Uncalibrated | 29 | 14 | 34 | 35 | 15 | 35 | 32 | 28 | 38 | 30 | 28 | 13 | **28** |
| | | Source-TS | 31 | 15 | 37 | 36 | 15 | 38 | 33 | 30 | 42 | 30 | 30 | 14 | 29 |
| | | PseudoCal | 29 | 14 | 34 | 35 | 15 | 35 | 33 | 28 | 38 | 31 | 28 | 14 | **28** |
| | | PLCC* | 33 | 16 | 36 | 37 | 15 | 35 | 33 | 28 | 39 | 32 | 30 | 16 | 29 |
| | | PLCC | 29 | 14 | 35 | 36 | 14 | 35 | 33 | 28 | 38 | 31 | 28 | 13 | **28** |

Table 17: 100 * BS on **DomainNet126**, using 15 bins (with the lowest in bold) across various SFDA classification tasks and methods with different calibration methods.

| SFDA | Type | Method | CR | CS | PC | PR | PS | RC | RS | SC | SR | Avg |
|---|---|---|---|---|---|---|---|---|---|---|---|---|
| DCPL | Oracle | CPCS | 32 | 60 | 52 | 29 | 62 | 44 | 58 | 41 | 32 | 46 |
| | | TransCal | 31 | 59 | 52 | 28 | 56 | 44 | 60 | 41 | 32 | 45 |
| | | UTDC* | 31 | 56 | 51 | 28 | 55 | 43 | 56 | 40 | 32 | 44 |
| | | Target-TS | 31 | 56 | 51 | 28 | 55 | 43 | 56 | 40 | 31 | 44 |
| | | Uncalibrated | 32 | 62 | 54 | 30 | 62 | 45 | 64 | 42 | 34 | 47 |
| | | Source-TS | 34 | 66 | 58 | 31 | 66 | 48 | 68 | 45 | 35 | 50 |
| | | NTS | 33 | 60 | 55 | 31 | 59 | 45 | 61 | 42 | 34 | 47 |
| | | PseudoCal | 31 | 56 | 51 | 28 | 55 | 43 | 57 | 40 | 31 | **44** |
| | | PLCC* | 33 | 56 | 52 | 29 | 58 | 43 | 57 | 41 | 33 | 45 |
| | | PLCC | 31 | 56 | 51 | 28 | 55 | 43 | 56 | 40 | 32 | **44** |
| SHOT | Oracle | CPCS | 37 | 59 | 63 | 34 | 66 | 50 | 60 | 44 | 36 | 50 |
| | | TransCal | 37 | 58 | 59 | 33 | 59 | 49 | 62 | 42 | 36 | 48 |
| | | UTDC* | 36 | 55 | 57 | 32 | 57 | 47 | 58 | 42 | 36 | 47 |
| | | Target-TS | 36 | 55 | 57 | 32 | 57 | 47 | 58 | 42 | 36 | 47 |
| | | Uncalibrated | 37 | 58 | 59 | 34 | 60 | 49 | 63 | 42 | 38 | 49 |
| | | Source-TS | 39 | 62 | 64 | 36 | 66 | 52 | 68 | 45 | 40 | 53 |
| | | PseudoCal | 36 | 55 | 57 | 32 | 57 | 47 | 58 | 42 | 36 | **47** |
| | | PLCC* | 36 | 55 | 57 | 32 | 58 | 47 | 58 | 42 | 37 | **47** |
| | | PLCC | 36 | 55 | 57 | 32 | 57 | 47 | 58 | 42 | 36 | **47** |
| AaD | Oracle | CPCS | 40 | 60 | 59 | 32 | 61 | 48 | 62 | 40 | 37 | 49 |
| | | TransCal | 39 | 61 | 60 | 31 | 58 | 48 | 62 | 40 | 37 | 48 |
| | | UTDC* | 39 | 57 | 58 | 31 | 57 | 47 | 60 | 40 | 37 | 47 |
| | | Target-TS | 39 | 57 | 58 | 31 | 57 | 47 | 60 | 40 | 37 | 47 |
| | | Uncalibrated | 41 | 63 | 63 | 32 | 63 | 49 | 68 | 41 | 39 | 51 |
| | | Source-TS | 43 | 67 | 68 | 34 | 68 | 53 | 72 | 44 | 41 | 54 |
| | | PseudoCal | 39 | 57 | 59 | 31 | 57 | 47 | 60 | 40 | 37 | **47** |
| | | PLCC* | 39 | 57 | 59 | 31 | 59 | 48 | 61 | 40 | 38 | 48 |
| | | PLCC | 39 | 57 | 58 | 31 | 58 | 47 | 61 | 40 | 37 | 48 |

### A.1.5    STATIC CALIBRATION ERROR RESULTS

Static Calibration Error (SCE) (Nixon et al., 2019), which is an extension of ECE to every probability in the multi class setting. SCE bins predictions separately for each class probability, computes the calibration error within the bin, and averages across bins. Note, unlike ECE, assuming infinite data and infinite bins, SCE is guaranteed to be zero if only if the model is calibrated. The formal definition of the SCE score is:

$$SCE = \frac{1}{K} \sum_{k=1}^{K} \sum_{m=1}^{M} \frac{n_{mk}}{n} |\text{acc}(m, k) - \text{conf}(m, k)| \tag{6}$$

Where, $\text{acc}(m, k)$ and $\text{conf}(m, k)$ are the accuracy and confidence of bin m for class label k, respectively; and $n_{mk}$ is the number of predictions in bin m for class label k;

Tables 18 and 19 present the SCE calibration results for VisDA and DomainNet126, respectively.

Table 18: 10,000 * SCE on **VisDA**, using 15 bins (with the lowest in bold) across various SFDA methods with different calibration methods.

| Type | Method | DCPL | SHOT | AaD |
|---|---|---|---|---|
| Oracle | CPCS | 13.39 | 15.93 | 12.86 |
| | TransCal | 13.28 | 16.60 | 12.91 |
| | UTDC* | 13.16 | 16.34 | 12.46 |
| | Target-TS | 13.00 | 16.34 | 12.46 |
| | Uncalibrated | 13.42 | 16.60 | 12.98 |
| | Source-TS | 13.16 | **16.40** | 12.95 |
| | NTS | **13.02** | N/A | N/A |
| | PseudoCal | 13.44 | 16.69 | **12.50** |
| | PLCC* | 16.74 | 19.13 | 15.17 |
| | PLCC | 13.13 | 16.47 | 12.52 |

Table 19: 10,000 * SCE on **DomainNet126**, using 15 bins (with the lowest in bold) across various SFDA classification tasks and methods with different calibration methods.

| SFDA | Type | Method | CR | CS | PC | PR | PS | RC | RS | SC | SR | Avg |
|---|---|---|---|---|---|---|---|---|---|---|---|---|
| DCPL | Oracle | CPCS | 0.19 | 0.46 | 0.41 | 0.17 | 0.39 | 0.32 | 0.38 | 0.33 | 0.19 | 0.32 |
| | | TransCal | 0.18 | 0.46 | 0.41 | 0.17 | 0.38 | 0.32 | 0.40 | 0.33 | 0.18 | 0.31 |
| | | UTDC* | 0.18 | 0.38 | 0.38 | 0.16 | 0.38 | 0.30 | 0.36 | 0.30 | 0.18 | 0.29 |
| | | Target-TS | 0.18 | 0.38 | 0.38 | 0.16 | 0.38 | 0.30 | 0.36 | 0.30 | 0.18 | 0.29 |
| | | Uncalibrated | 0.19 | 0.48 | 0.43 | 0.17 | 0.47 | 0.33 | 0.43 | 0.35 | 0.20 | 0.34 |
| | | Source-TS | 0.20 | 0.48 | 0.44 | 0.17 | 0.48 | 0.33 | 0.44 | 0.36 | 0.20 | 0.34 |
| | | NTS | 0.20 | 0.46 | 0.43 | 0.17 | 0.45 | 0.33 | 0.41 | 0.34 | 0.20 | 0.33 |
| | | PseudoCal | 0.18 | 0.39 | 0.39 | 0.16 | 0.39 | 0.3 | 0.35 | 0.31 | 0.18 | **0.29** |
| | | PLCC* | 0.18 | 0.37 | 0.39 | 0.16 | 0.37 | 0.30 | 0.35 | 0.30 | 0.18 | **0.29** |
| | | PLCC | 0.18 | 0.38 | 0.39 | 0.17 | 0.38 | 0.30 | 0.36 | 0.30 | 0.18 | **0.29** |
| SHOT | Oracle | CPCS | 0.23 | 0.46 | 0.53 | 0.20 | 0.45 | 0.39 | 0.42 | 0.35 | 0.22 | 0.36 |
| | | TransCal | 0.23 | 0.45 | 0.50 | 0.20 | 0.45 | 0.38 | 0.44 | 0.34 | 0.21 | 0.35 |
| | | UTDC* | 0.21 | 0.38 | 0.46 | 0.18 | 0.41 | 0.34 | 0.37 | 0.33 | 0.21 | 0.32 |
| | | Target-TS | 0.21 | 0.38 | 0.46 | 0.18 | 0.41 | 0.34 | 0.37 | 0.31 | 0.21 | 0.32 |
| | | Uncalibrated | 0.23 | 0.45 | 0.50 | 0.20 | 0.48 | 0.38 | 0.45 | 0.34 | 0.23 | 0.36 |
| | | Source-TS | 0.23 | 0.48 | 0.54 | 0.20 | 0.52 | 0.41 | 0.48 | 0.38 | 0.23 | 0.39 |
| | | PseudoCal | 0.21 | 0.39 | 0.46 | 0.19 | 0.42 | 0.34 | 0.37 | 0.31 | 0.21 | 0.32 |
| | | PLCC* | 0.21 | 0.37 | 0.45 | 0.18 | 0.39 | 0.34 | 0.37 | 0.30 | 0.20 | **0.31** |
| | | PLCC | 0.21 | 0.38 | 0.46 | 0.19 | 0.41 | 0.34 | 0.37 | 0.31 | 0.21 | 0.32 |
| AaD | Oracle | CPCS | 0.28 | 0.48 | 0.58 | 0.20 | 0.46 | 0.41 | 0.43 | 0.35 | 0.24 | 0.38 |
| | | TransCal | 0.26 | 0.50 | 0.59 | 0.20 | 0.44 | 0.41 | 0.43 | 0.35 | 0.23 | 0.38 |
| | | UTDC* | 0.25 | 0.41 | 0.52 | 0.19 | 0.43 | 0.38 | 0.40 | 0.31 | 0.24 | 0.35 |
| | | Target-TS | 0.25 | 0.40 | 0.53 | 0.19 | 0.43 | 0.38 | 0.39 | 0.31 | 0.24 | 0.35 |
| | | Uncalibrated | 0.28 | 0.51 | 0.63 | 0.20 | 0.55 | 0.43 | 0.49 | 0.35 | 0.27 | 0.41 |
| | | Source-TS | 0.28 | 0.54 | 0.67 | 0.21 | 0.58 | 0.46 | 0.52 | 0.38 | 0.26 | 0.43 |
| | | PseudoCal | 0.26 | 0.43 | 0.56 | 0.19 | 0.45 | 0.37 | 0.39 | 0.31 | 0.24 | 0.36 |
| | | PLCC* | 0.25 | 0.39 | 0.50 | 0.19 | 0.39 | 0.37 | 0.38 | 0.29 | 0.23 | **0.33** |
| | | PLCC | 0.25 | 0.39 | 0.52 | 0.19 | 0.41 | 0.37 | 0.38 | 0.29 | 0.24 | 0.34 |

## A.2 MODELS ACCURACY

Tables 20, 21, 22, and 23 provide the accuracy levels are associated with the displayed calibration levels for Office-Home, VisDA, DomainNet40, and DomainNet126, respectively.

Table 20: Accuracy on **Office-Home**, across various SFDA classification tasks and methods

| SFDA | AC | AP | AR | CA | CP | CR | PA | PC | PR | RA | RC | RP | Avg |
|------|------|------|------|------|------|------|------|------|------|------|------|------|------|
| DCPL | 66.1 | 83.3 | 84.5 | 72.4 | 84.5 | 82.6 | 72.7 | 62.4 | 83.5 | 75.0 | 66.4 | 87.6 | 76.7 |
| SHOT | 55.5 | 77.2 | 79.4 | 66.3 | 76.9 | 75.9 | 67.4 | 53.5 | 80.4 | 72.5 | 58.2 | 83.2 | 70.5 |
| AAD | 56.0 | 77.4 | 77.6 | 64.4 | 75.6 | 75.1 | 62.4 | 55.4 | 77.8 | 68.8 | 57.5 | 82.8 | 69.2 |

Table 21: Accuracy on **VisDA**, across various SFDA classification tasks and methods

| SFDA | SR |
|------|------|
| DCPL | 82.4 |
| SHOT | 78.7 |
| AAD | 84.9 |

Table 22: Accuracy on **DomainNet40**, across various SFDA classification tasks and methods

| SFDA | CP | CR | CS | PC | PR | PS | RC | RP | RS | SC | SP | SR | Avg |
|------|------|------|------|------|------|------|------|------|------|------|------|------|------|
| DCPL | 81.0 | 92.4 | 76.7 | 77.0 | 91.1 | 75.4 | 79.8 | 81.9 | 73.0 | 80.1 | 80.7 | 90.2 | 81.6 |
| SHOT | 78.7 | 90.3 | 74.2 | 72.0 | 88.2 | 74.1 | 78.1 | 81.0 | 72.7 | 77.4 | 79.7 | 88.6 | 79.6 |
| AAD | 79.5 | 91.0 | 76.1 | 75.1 | 90.5 | 75.1 | 77.8 | 80. | 73.3 | 79.7 | 81.3 | 91.4 | 80.9 |

Table 23: Accuracy on **DomainNet126**, across various SFDA classification tasks and methods

| SFDA | CR | CS | PC | PR | PS | RC | RS | SC | SR | Avg |
|------|------|------|------|------|------|------|------|------|------|------|
| DCPL | 80.8 | 62.0 | 65.4 | 82.3 | 61.8 | 71.6 | 61.5 | 73.2 | 80.4 | 71.0 |
| SHOT | 77.0 | 61.0 | 59.8 | 79.0 | 58.8 | 67.9 | 59.1 | 71.1 | 76.3 | 67.8 |
| AAD | 75.9 | 59.6 | 59.0 | 80.5 | 58.9 | 68.4 | 57.8 | 72.6 | 77.1 | 67.8 |

## A.3 CALIBRATION COMPARATIVE RESULTS AS A FUNCTION OF THE NUMBER OF BINS

We confirmed that the choice of bin count for calibration has no effect on the performance of our method. Testing with bin counts ranging from 3 to 21 revealed consistent PLCC results, as demonstrated in Fig. 5.

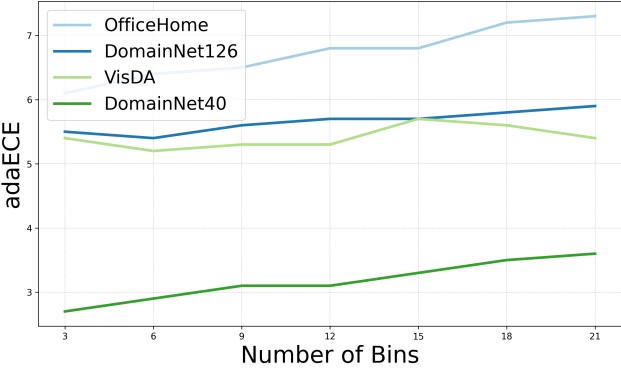

Figure 5: The adaECE results for the PLCC method were evaluated across multiple datasets with varying bin numbers. Both calibration and evaluation processes utilized the adaECE metric.

## A.4    SUBSET OF TRUE LABELS

In scenarios where labeling target domain data is inexpensive, one might consider labeling a small portion of the data for calibration purposes. However, in some instances, a small labeled subset may not be sufficient, or the target data itself may be limited. Fig. 6 demonstrates the proportion of the target domain validation set required to surpass the calibration performance of PLCC when using a subset of true label examples. The results indicate that in certain cases, labeling over 50% of the validation set may be necessary.

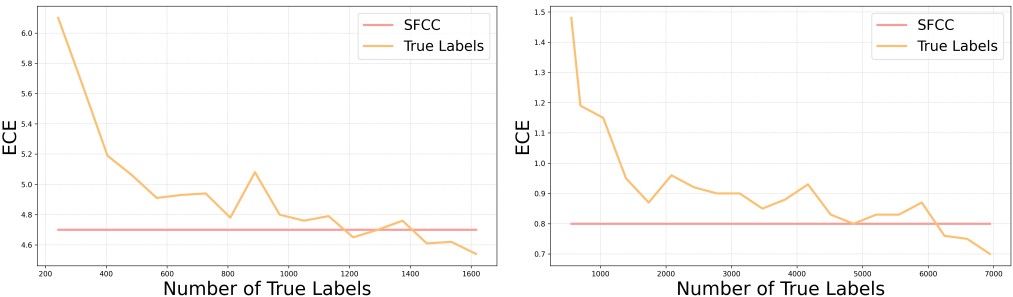

Figure 6: Comparison of ECE loss between PLCC and temperature scaling applied to different proportions of labeled target examples. Dataset: **DomainNet40**, SFDA: DCPL, number of bins: 15

