# OpenReview forum: "Confidence Calibration in Source-Free Domain Adaptation based on Pseudo-Labels"
_ICLR.cc/2026/Conference — Submitted to ICLR 2026_

### Official Review · Reviewer_oQVS · 2025-10-26

**Soundness:** 2
**Presentation:** 2
**Contribution:** 2
**Rating:** 4
**Confidence:** 3

**Summary:**

This paper addresses the challenge of confidence calibration in Source-Free Domain Adaptation (SFDA), a setting where pre-trained source models are adapted to unlabeled target domains without access to source data (due to privacy, legal, or proprietary constraints). A key limitation of existing work is that most calibration methods rely on labeled source data or labeled target data, i.e., assumptions incompatible with SFDA. The paper’s core empirical insight is that noisy pseudo-labels (generated from the source model) can reliably approximate true labels for accuracy estimation: even with significant noise, the accuracy computed using pseudo-labels closely matches that of true labels. Building on this, the authors propose Pseudo-Label Confidence Calibration (PLCC), a lightweight algorithm that: (1) generates Enhanced Pseudo-Labels (EPL) using a pre-trained feature extractor (Swin-B) and class centroids (via cosine distance), (2) estimates bin-wise accuracy using EPL, and (3) applies temperature scaling to minimize Adaptive Expected Calibration Error (adaECE) on the target domain.

**Strengths:**

1. SFDA is critical for real-world deployments (e.g., medical imaging, autonomous driving) where source data access is restricted. The paper fills a key gap by focusing on confidence calibration.
2. The core insight (pseudo-labels approximate true labels for accuracy estimation) is rigorously validated.
3. Tests on 4 standard benchmarks and 3 distinct SFDA methods (DCPL, SHOT, AaD) ensure generalizability.

**Weaknesses:**

1. The paper’s core insight (pseudo-label accuracy approximates true accuracy) is supported empirically but lacks theoretical grounding. For example, no formal proof or bounds on the error between pseudo-label-based accuracy and true accuracy.
2. Minor typos (e.g., "AAD" instead of "AaD" in Tables 21–23) distract from readability.
3. No comparison to recent SFDA calibration methods (e.g., post-2023 work beyond PseudoCal) to contextualize PLCC’s state-of-the-art claims.

**Questions:**

1. How centroids are initialized and updated during adaptation (e.g., are centroids fixed after initial computation, or refined iteratively?).
2. Why Swin-B is chosen over other feature extractors (e.g., ViT, ResNet), no ablation on feature extractor impact.

---

> ### Author Response · Authors · 2025-11-17
>
> We thank the reviewer for the comments.
>
> W1: Eq. 3 states that pseudo-labels, although very noisy, can still provide a good approximation of the adapted model’s accuracy. Eq. 3 is not a theoretically guaranteed property of pseudo-labels; rather, it reflects a surprisingly consistent empirical behavior that motivates the design of our algorithm and has been validated across a wide range of domain-shift datasets. We emphasize that the lack of theoretical grounding is not due to mathematical intractability, but to the absence of a clear parametric model that can formally capture complex real-world phenomena such as domain shift and pseudo-label noise.
>
> Although we acknowledge the absence of a formal theoretical proof or bounds on the discrepancy between pseudo-label and true accuracy, this situation aligns with the fact that current methods in confidence calibration, domain adaptation, and source-free accuracy estimation are supported primarily by empirical evidence rather than formal theory. For instance, the widely adopted Temperature Scaling method and the Expected Calibration Error (ECE) metric were both introduced without formal theoretical justification, yet have become standard tools due to their strong empirical performance.
>
> W3: We have included comparisons with all SFDA calibration methods we are aware of. If there are others we may have missed, we would appreciate the reviewer’s suggestions and will gladly add them.
>
>
> Q1: The centroids are fixed after their initial computation. We experimented with updating them at every iteration; however, this did not improve calibration performance.
> Q2: We evaluated both SWIN-B and ResNet architectures and observed that SWIN-B consistently achieved superior results overall.

---

> > ### Comment · Reviewer_oQVS · 2025-11-26
> >
> > Thanks for the authors’ response. Overall, the paper is somewhat narrow in scope, focusing mainly on the calibration of source-free domain adaptation. Moreover, the calibration with pseudo labels lacks solid theoretical grounding, which weakens the significance of the work.
> >
> > The authors state that they evaluated both SWIN-B and ResNet architectures and observed that SWIN-B consistently achieved better performance. However, they do not provide any concrete comparative results to support this claim.
> >
> > Given the overall quality of the paper and the comments from other reviewers, I tend to maintain my current rating.

---

### Official Review · Reviewer_K3qN · 2025-10-27

**Soundness:** 2
**Presentation:** 1
**Contribution:** 2
**Rating:** 2
**Confidence:** 4

**Summary:**

This paper introduces PLCC, a method for calibrating models in source-free domain adaptation using pseudo labels instead of true labels. It assumes that the agreement between model predictions and pseudo labels approximates their agreement with ground truth, enabling label-free calibration estimation. The authors enhance pseudo labels through an EPL process that refines class assignments using pre-trained features and class centroids. Experiments on several benchmarks show that PLCC achieves lower calibration errors compared to existing methods.

**Strengths:**

1. The paper tackles an important and practical problem, i.e., calibration for source-free domain adaptation, which has received limited attention.
2. The proposed PLCC method is conceptually simple and easy to implement.
3. The authors conduct extensive experiments on multiple benchmarks, showing consistent improvements in calibration metrics.
4. The paper is well organized and reproducible.

**Weaknesses:**

1. I think the main weakness of this paper is that it approaches the problem based on confidence calibration theory, but the theoretical depth is far from enough. This needs to be strengthened. Specifically, the conditions and implications of equations (3) and (4) are worth calibrating and considering. In addition, the paper relies on a marginal approximation rather than a conditionally valid equality, which the authors describe as a “very mild assumption.” However, the manuscript does not specify under what distributions or scenarios this assumption holds or provide any theoretical limits on when it fails. Moreover, while Equation (5) approximates overall accuracy using pseudo labels, there is no analysis of the estimation error’s bias, variance, or sample complexity, nor any bound under varying noise rates. This weakens the theoretical soundness of the proposed approach.
2. No ablation on pseudo-label quality. The method assumes EPL substantially improves label reliability, but there is no quantitative study of pseudo-label noise rate before and after EPL.
3. The approach is designed specifically for closed-set classification; it is unclear whether PLCC works for open-set or multi-label tasks common in SFDA research.
4. Lack of UDA task-specific qualitative analysis.

**Questions:**

The methodology in this paper appears relatively simple, suggesting that the author could strengthen it with more in-depth theoretical foundations and sound analytical support to demonstrate its effectiveness. I believe the current version of the paper is somewhat superficial and could benefit from greater theoretical depth. In addition, the experimental design may warrant further consideration.

---

> ### Author Response · Authors · 2025-11-17
>
> We thank the reviewer for the comments.
>
> Eq. 3 states that pseudo-labels, although very noisy, can still provide a good approximation of the adapted model’s accuracy. Eq. 3 is not a theoretically guaranteed property of pseudo-labels; rather, it reflects a surprisingly consistent empirical behavior that motivates the design of our algorithm and has been validated across a wide range of domain-shift datasets. We emphasize that the lack of theoretical grounding is not due to mathematical intractability, but to the absence of a clear parametric model that can formally capture complex real-world phenomena such as domain shift and pseudo-label noise.
>
> Many deep learning algorithms lack formal theoretical guarantees and are primarily validated through empirical performance improvements over prior state-of-the-art (SOTA) methods. This is especially true for confidence calibration and domain adaptation in general, and even more so for source-free calibration and accuracy estimation. All SOTA approaches compared in our paper were originally justified by their empirical success rather than theoretical proofs. Notably, even widely adopted methods such as Temperature Scaling and the Expected Calibration Error (ECE) metric were introduced without formal theoretical justification.
>
> Furthermore, Fig 3 presents the empirical distribution corresponding to Equation 5 across all datasets evaluated in our study. From this figure, one can directly infer the estimation bias (0.86) and the variance of the error. Regarding sample complexity, we rely on real-world datasets and observe strong performance even when adapting models with only a few hundred samples.
>
> 2. We show in  Figure 2a that the EPL method effectively reduces pseudo-label noise by approximately 50\% on average.
> In all our experiments, we show results on a variant of PLCC (denoted PLCC*) where we use PL instead of EPL.
>
> 3. Our work specifically focuses on the source-free domain adaptation (SFDA) calibration problem in the closed-set setting. To the best of our knowledge, there is currently no established solution addressing this specific problem. While extending the method to more complex scenarios is an interesting direction for future work, it lies beyond the primary scope of this article.
>
> 4. We are not entirely certain which aspect the reviewer is referring to. What kind of qualitative analysis are you expecting to see?

---

> > ### Comment · Reviewer_K3qN · 2025-11-22
> >
> > Thank you for the authors’ response. However, I believe that the theoretical and experimental limitations of the paper have not yet been adequately addressed. Specifically, given that the authors propose a “very mild assumption,” a more comprehensive justification is necessary to meet ICLR’s standards for theoretical soundness. While it is true, as the authors note, that many deep learning algorithms lack theoretical guarantees, this does not in itself constitute a sufficient reason to forgo a more rigorous analysis.
> >
> > Furthermore, the experimental evaluation has not been substantially improved, and its limitations remain concerning. In particular, the baseline methods used for comparison appear somewhat outdated and not sufficiently representative. Additionally, the absence of case analyses illustrating how the proposed mechanism leads to performance gains makes it difficult for readers to fully understand its contribution.

---

### Official Review · Reviewer_toFH · 2025-10-28

**Soundness:** 2
**Presentation:** 2
**Contribution:** 2
**Rating:** 4
**Confidence:** 3

**Summary:**

This paper proposes a source-free confidence calibration method. Through experimental observations, the authors find that noisy pseudo-labels can serve as effective surrogates for true labels in source-free domain adaptation calibration, even without access to target ground-truth labels. Building on this insight, they introduce PLCC, a prototype- and enhanced pseudo-label–based confidence calibration approach that derives a final temperature scale for model calibration. Experimental results demonstrate the effectiveness of the proposed method.

**Strengths:**

- The problem studied in this paper is meaningful and challenging.

- The proposed method has been validated on standard SFDA benchmarks and shows effectiveness.

**Weaknesses:**

- One major concern lies in the strong assumption behind Equation (3) in the main paper. Is this equality a general phenomenon? Since it serves as the key motivation for algorithm development, clarifying this assumption is crucial to ensuring the soundness of the paper.

- If noisy pseudo-labels can already be regarded as true labels for SFDA calibration, it remains unclear why the paper introduces the enhanced pseudo-label strategy.

- It appears that most of the performance gains stem from the enhanced pseudo-labels, which, however, are not thoroughly discussed. As this enhancement depends on an external pre-trained model, have the authors explored alternative pseudo-label enhancement strategies?

- When discussing computational complexity, is the use of the external pre-trained model taken into account?

- The overall presentation could be further improved. Please refer to the detailed comments below:

    - In the abstract, it is unclear what accuracy refers to — is it a pure performance metric or a calibration-related measure?

    - The methodology section could highlight the key points more clearly. If the main technical novelty lies primarily in temperature scaling, it may be considered somewhat limited.

    - Minor suggestion: Since the method is developed based on empirical observations, Figure 4 would be more appropriately placed earlier in the paper (e.g., in the Introduction) to better motivate the approach.

- There is a lack of the LLM usage declaration in the manuscript.

**Questions:**

I have listed all my questions and concerns in the Weaknesses section.

---

> ### Author Response · Authors · 2025-11-17
>
> We thank the reviewer for the comments.
>
>
> 1. Eq. 3 is not a theoretically guaranteed property of pseudo-labels. Rather, it  describes a surprising behavior of pseudo-labels that motivates our algorithm and is empirically validated on a wide range of domain-shift datasets. It is important to note that the concern here is not the difficulty of finding a mathematical proof, but rather the absence of a clear parametric mathematical model for concepts such as domain shift and pseudo-label noise. We note that many deep learning algorithms lack a theoretical foundation and are instead validated by demonstrating superior results compared to previous SOTA methods. This is particularly true for confidence calibration and domain adaptation in general, and for source-free calibration and accuracy estimation in particular. All the SOTA methods we compare in our paper were originally justified based on their improved results. Even the Temperature Scaling calibration method  and the ECE calibration measure were introduced without a theoretical justification.
>
> 2. While noisy pseudo-labels can sometimes yield reasonable calibration results, our Enhanced Pseudo-Labels (EPL) method achieves substantially better performance. To illustrate this, we included  PLCC*—a variant of our approach that uses standard pseudo-labels without the enhancement process—as an ablation study in our paper. We show in  Figure 2a that the EPL method effectively reduces pseudo-label noise by approximately 50\% on average.
>
> 3. The EPL results reported in the paper were obtained using the SWIN-B backbone. We also experimented with ResNet, but observed that SWIN-B  produced overall better calibration results.
>
> 4. Our approach does not require model training, making it computationally efficient. EPL generation can be performed on a standard GPU, and the calibration results are obtained almost instantaneously.
>
> 5,6. We appreciate the reviewer’s feedback and will incorporate the suggested changes into the final version of the paper.

---

### Meta-Review · Area_Chair_DaAC · 2026-01-06

**Summary:**

All three reviewers raise concerns about both the theoretical soundness and empirical support of the paper. As indicated by all of them the core motivation relies on a strong and insufficiently justified assumption in Equation (3), treated as a “very mild assumption” despite lacking conditions, validity ranges, or failure cases, which undermines the confidence calibration theory underlying the method. If noisy pseudo-labels are already assumed to approximate true labels for SFDA calibration, the necessity and role of the enhanced pseudo-label (EPL) strategy are unclear, especially since most performance gains appear to stem from this enhancement, which depends on an external pre-trained model whose computational cost and alternatives are not discussed or ablated. The theoretical analysis is shallow, providing no bounds or bias–variance analysis for pseudo-label-based accuracy estimation, relying on marginal approximations without justification, and offering no quantitative study of pseudo-label noise reduction. Empirically, key aspects are missing, including ablations on pseudo-label quality, comparisons to recent SFDA calibration methods, and qualitative analyses tailored to UDA tasks, while the method’s applicability beyond closed-set classification remains unclear.

**Reviewer Concerns:**

While the discussion process has been limited two of the reviewers have already read the provided rebuttal of the authors and have commented further of this. They partially agree with the answers of the reviewers but they also state once again that they want to maintain their original scores. The main reason is that the theoretical and experimental limitations of the paper have not been adequately addressed and the experimental results are not fully convincing. The authors have tried in their rebuttal to address the concern regarding Eq. 3 but they did not provide too many details. Also, they have not commented back to the replies of the two reviewers.

**Reviewer Scores:**

I think the discussion might have improved some of the opinions of the reviewers but I doubt the overall outcome would have been different. The main problem is that all the initial scores were rather negative and the concerns were real. This has been acknowledged also by the authors in their rebuttal. Overall, I think the authors could have done a better job in their rebuttal and should have insisted more on the main issues of the manuscript as highlighted by all three reviewers.

---

### Decision · Program_Chairs · 2026-01-26

Reject